# Better Training Data Attribution via Better Inverse Hessian-Vector Products

**Andrew Wang**[*1,2]    **Elisa Nguyen**[3]    **Runshi Yang**[1,2]
**Juhan Bae**[1,2]    **Sheila A. McIlraith**[1,2,4]    **Roger Grosse**[1,2,4]
[1]University of Toronto    [2]Vector Institute for Artificial Intelligence
[3]Tübingen AI Center, University of Tübingen
[4]Schwartz Reisman Institute for Technology and Society

## Abstract

Training data attribution (TDA) provides insights into which training data is responsible for a learned model behavior. Gradient-based TDA methods such as influence functions and unrolled differentiation both involve a computation that resembles an inverse Hessian-vector product (iHVP), which is difficult to approximate efficiently. We introduce an algorithm (ASTRA) which uses the EKFAC-preconditioner on Neumann series iterations to arrive at an accurate iHVP approximation for TDA. ASTRA is easy to tune, requires fewer iterations than Neumann series iterations, and is more accurate than EKFAC-based approximations. Using ASTRA, we show that improving the accuracy of the iHVP approximation can significantly improve TDA performance.

## 1   Introduction

Machine learning systems derive their behavior from the data they are trained on. *Training data attribution* (TDA) is a family of techniques that help uncover how individual training examples influence model predictions. As such, TDA is a valuable tool with applications in data valuation and curation [1–4], interpreting model behavior [5–11], building more equitable and transparent machine learning systems [12, 13] and investigating questions of intellectual property and copyright by tracing outputs back to specific data sources [11, 14, 15], among other applications.

Influence functions (IF) [16, 17] and unrolled differentiation [4, 18–21] are two gradient-based TDA methods that involve, or can be approximated as computing inverse Hessian-vector products (iHVPs).[1] Inverting the Hessian is infeasible but for the smallest of neural networks, so the iHVP is typically computed without explicitly constructing the Hessian. There are a number of choices available: Koh and Liang [17], who first introduced influence functions to deep learning, use the iterative algorithm LiSSA [23], which is a method based on stochastic Neumann series iterations[2] (SNI) [24, 25] that can take *thousands* of iterations to converge to an unbiased solution [7, 17]. Alternatively, Grosse et al. [7] adopt a tractable parametric approximation to the Hessian [1, 11, 19] using Eigenvalue-corrected Kronecker Factorization (EKFAC) [26, 27]. EKFAC makes several simplifying assumptions that hold only approximately in practice, but they dramatically lower both computational and memory cost, making it feasible to scale to billion-parameter language models [7]. However, EKFAC influence functions (EKFAC-IF) computed on converged models only correlate modestly with ground truth

---

[*]Correspondence to andrewwang@cs.toronto.edu.

[1]Other gradient-based TDA methods such as TRAK [22] or LOGRA [1] also involve iHVPs but use additional techniques such as random/PCA gradient projection.

[2]LiSSA was introduced in the context of optimization and contains other components, but we refer to the component that computes the iHVP, as found in the Koh and Liang [17] implementation. Henceforth, we will use the terms LiSSA and SNI interchangeably, with their slight difference described in Appendix B.

39th Conference on Neural Information Processing Systems (NeurIPS 2025).

over a variety of datasets and model architectures [19] and tend to struggle for architectures involving convolution, suggesting further room for improvement. We aim to improve EKFAC-based TDA methods in this paper by improving the iHVP approximation.

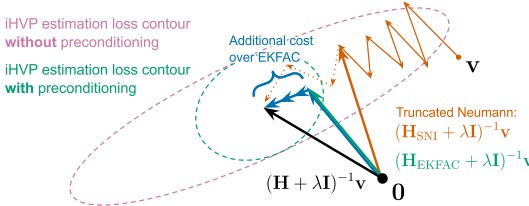

Figure 1: The objective is to compute the damped iHVP $(\mathbf{H} + \lambda\mathbf{I})^{-1}\mathbf{v}$. Preconditioning Stochastic Neumann Iterations (SNI) with EKFAC (ASTRA) improves the convergence speed of the iHVP approximation. Initialized at $\mathbf{0}$, it results in the same approximation as using EKFAC after one iteration. SNI may require thousands of iterations to converge, and truncating early results in undesirable implicit damping.

Computing iHVPs in the context of TDA can be seen as finding the minimizer to high-dimensional quadratic optimization problems in parameter space [25, 28, 29]. Curvature matrices for large neural networks are known to be ill-conditioned [30, 31], causing slow convergence for iterative methods such as SNI. While costly and rather difficult to tune [17, 32, 33], the upside of SNI is that it produces a *consistent* estimator – the algorithm converges to the iHVP in the limit as more compute is used [23]. In contrast, while EKFAC often provides a better cost vs. accuracy tradeoff for TDA [7], there is no simple way to improve its accuracy by applying more compute. Our central insight is that we can combine the best of both worlds: we can repurpose the EKFAC decomposition — which needs to be computed for EKFAC-based influence functions and unrolled differentiation anyways — as a preconditioner for SNI, yielding a cost effective procedure for improving the iHVP accuracy (Figure 1). Our contributions are as follows:

First, **we present an algorithm called ASTRA which uses EKFAC as a preconditioner for SNI with the aim of computing cost-effective and accurate iHVPs**. ASTRA can be applied to both influence functions (ASTRA-IF) and unrolled differentiation via an approximation called SOURCE (ASTRA-SOURCE) [19] among other applications. To the best of our knowledge, no prior work has applied the EKFAC preconditioner to SNI for the TDA setting. In our experiments, the incremental cost of ASTRA-IF and ASTRA-SOURCE was only *hundreds* of iterations, compared to EKFAC-IF and EKFAC-SOURCE, respectively. In contrast to other settings in which we would encounter iHVPs, we are often willing to pay this extra computational cost to obtain a more accurate iHVP for TDA.

Second, while past papers have questioned the reliability of influence functions [34, 35], **we show that influence functions computed with accurate iHVPs have strong predictive power**, even in settings in which the assumptions in its derivations may not hold [35, 36]. In our experiments, ASTRA-IF was able to achieve a Spearman correlation score [37] with ground-truth retraining of around 0.5 across many settings,[3] and ensembling these predictions often raised performance to 0.6. ASTRA provides an accurate approximation of the iHVP, which significantly increases the efficacy of ensembling [19, 22] compared to EKFAC and also significantly improves TDA performance for architectures involving convolution layers. For the experiment involving a convolution architecture, performance of ensembled ASTRA-IF was 0.6, a large increase from EKFAC-IF's 0.25.

Finally, **leveraging EKFAC's eigendecomposition, we show that low curvature directions are essential for high quality influence estimates**. Truncating Neumann series has an implicit damping effect [29, 38], which has a disproportional impact on low curvature directions. To quantify the downstream impact on influence functions performance, we perform an ablation study by using the EKFAC eigendecomposition to project the iHVP onto subspaces containing different levels of curvature during Neumann series iterations. We show that these low curvature components are essential for high quality influence estimates. This suggests that good influence function performance demands careful hyperparameter tuning, which can be costly, especially for off-the-shelf iterative solvers. In contrast, ASTRA requires much less hyperparameter tuning; we use a set of hyperparameters determined by simple heuristics which worked well for all of our experiments. We also note that while EKFAC was introduced as a preconditioner in second-order optimization [26], its compact representation of the eigendecomposition remains relatively underappreciated — the method in this ablation study in which we use the EKFAC eigendecomposition to analyze the quadratic objective could therefore be of independent interest.

---

[3]More precisely, we use a widely-used evaluation metric called Linear Datamodeling Score (LDS) [22], which measures the rank correlation between ground-truth retraining outcomes and a TDA algorithm's predictions. LDS is defined and performance comparisons are provided in Section 5.

## 2 Preliminaries

This section briefly introduces preliminaries relating to 1) computing the iHVP and 2) influence functions. To help the reader navigate the mathematical objects, we have provided a table of notations and acronyms in Appendix A. Further background and expanded discussion is provided in Appendix B.

Given a training dataset $\mathcal{D} = \{z_i\}_{i=1}^N$ where $z_i = (\mathbf{x}^{(i)}, \mathbf{t}^{(i)})$ is an input-target pair, and a model parameterized by $\boldsymbol{\theta} \in \mathbb{R}^D$, let $g(\boldsymbol{\theta}, \mathbf{x}^{(i)})$ denote the model output on $\mathbf{x}^{(i)}$, let $\mathcal{L}(\mathbf{y}, \mathbf{t})$ be a convex loss function. We define a training objective $\mathcal{J}(\boldsymbol{\theta}, \mathcal{D}) := \frac{1}{N} \sum_{i=1}^N \mathcal{L}(g(\boldsymbol{\theta}, \mathbf{x}^{(i)}), \mathbf{t}^{(i)})$ as the average loss over $\mathcal{D}$. Given a query data point $z_q$ and a measurement function $f_{z_q}(\boldsymbol{\theta})$, such as correct-class margin [22], an *idealized* objective of TDA is to approximate the impact of removing a training example $z_m$ from the training dataset $\mathcal{D}$ on $f_{z_q}$. A pointwise TDA method $\tau$ assigns a score $\tau(z_m, z_q, \mathcal{D})$ that measures the impact of removing $z_m$ from $\mathcal{D}$ on $f_{z_q}$. Since modern neural networks often exhibit multiple optima, the stochasticity in the optimization process from sources such as parameter initialization [39], sampled dropout masks [40], and mini-batch ordering [41] can result in different learned optima, which we denote $\boldsymbol{\theta}^s$. Let $\xi$ be a random variable which captures this stochasticity in the training procedure [42, 43].

### 2.1 Computing Inverse Hessian-Vector Products for TDA

Inverse Hessian-vector products (iHVPs) are ubiquitous in many machine learning settings beyond TDA [25, 26, 44, 45], such as second-order optimization [26, 46, 47], but different settings have different considerations when trading off cost vs. accuracy. The canonical second-order optimization method – Newton's method – utilizes an inverse Hessian-gradient product to compute the Newton step, and may achieve faster local convergence than first-order methods [28]. However, computing an iHVP is substantially more expensive than a standard gradient computation. In optimization, there exists a tradeoff between devoting extra compute to obtain a better iHVP approximation versus taking more steps in the optimization procedure [48]. Because most deep learning optimizers rely on stochastic updates, the extra cost of a highly precise curvature estimate often is not justified and popular optimizers such as Adam [47] default to the much cheaper diagonal preconditioner. In contrast, we are typically willing to pay a higher cost for accurate iHVP approximations in the TDA context, since – as we will show – an accurate iHVP may significantly improve TDA performance. Two popular ways of computing the iHVP for TDA are EKFAC [7, 26, 27] and LiSSA [17, 23]. We now briefly describe each method.

**Computing the iHVP with EKFAC** KFAC [26] and EKFAC [27] make a block-diagonal parametric approximation of the Fisher Information Matrix[4] (FIM) so that its eigendecomposition can be done for each layer $l$ independently, which is significantly cheaper than an often infeasible brute-force eigendecomposition. The FIM is defined as:

$$\mathbf{F} := \frac{1}{N} \sum_{\mathbf{x}^{(i)} \in \mathcal{D}} \mathbb{E}_{\hat{\mathbf{y}} \sim p_{\boldsymbol{\theta}}(\mathbf{y}|\mathbf{x}^{(i)})} \left[ \nabla_{\boldsymbol{\theta}} \log p_{\boldsymbol{\theta}}(\hat{\mathbf{y}}|\mathbf{x}^{(i)}) \nabla_{\boldsymbol{\theta}} \log p_{\boldsymbol{\theta}}(\hat{\mathbf{y}}|\mathbf{x}^{(i)})^\top \right], \tag{1}$$

where $p_{\boldsymbol{\theta}}(\mathbf{y}|\mathbf{x})$ is the model's own distribution over targets. For multi-layer perceptrons, KFAC approximates the $l$th block of the FIM with:

$$\mathbf{F}_l \approx \underbrace{\mathbb{E}\left[\bar{\mathbf{a}}_{l-1} \bar{\mathbf{a}}_{l-1}^\top\right]}_{\mathbf{A}_{l-1}} \otimes \underbrace{\mathbb{E}\left[\mathcal{D}\mathbf{s}_l \mathcal{D}\mathbf{s}_l^\top\right]}_{\mathbf{S}_l}, \tag{2}$$

$$= (\mathbf{Q}_{\mathbf{A}_{l-1}} \otimes \mathbf{Q}_{\mathbf{S}_l}) \underbrace{(\mathbf{D}_{\mathbf{A}_{l-1}} \otimes \mathbf{D}_{\mathbf{S}_l})}_{\mathbf{\Lambda}_{l,\text{KFAC}}} (\mathbf{Q}_{\mathbf{A}_{l-1}} \otimes \mathbf{Q}_{\mathbf{S}_l})^\top \tag{3}$$

where $\bar{\mathbf{a}}_{l-1}$ are the activations of the $l-1$th layer[5] and $\mathcal{D}\mathbf{s}_l$ are the pseudogradients[6] of the loss with respect to the preactivations $\mathbf{S}_l$ of the $l$th layer, and $\mathbf{A}_{l-1} = \mathbf{Q}_{\mathbf{A}_{l-1}} \mathbf{D}_{\mathbf{A}_{l-1}} \mathbf{Q}_{\mathbf{A}_{l-1}}^\top$ and $\mathbf{S}_l =$

---

[4]For standard regression and classification tasks whose outputs can be seen as the natural parameters of an exponential family, the Fisher Information Matrix $\mathbf{F}$ and the Generalized Gauss-Newton Hessian $\mathbf{G}$ coincide, which we will use as a substitute for the Hessian $\mathbf{H}$. For details, see Appendix B.

[5]The bar indicates the use of the homogeneous vector notation $\bar{\mathbf{a}}_{l-1} = \begin{bmatrix} \mathbf{a}_{l-1}^\top & 1 \end{bmatrix}^\top$.

[6]By pseudogradients, we mean the gradient of the loss using a sampled target with respect to the parameters.

$\mathbf{Q}_{\mathbf{S}_l}\mathbf{D}_{\mathbf{S}_l}\mathbf{Q}_{\mathbf{S}_l}^\top$ are eigendecompositions of $\mathbf{A}_{l-1}$ and $\mathbf{S}_l$ respectively. We denote the block-diagonal matrix approximated this way as $\mathbf{F}_{\text{KFAC}}$. We can then approximate the inverse FIM as $(\mathbf{F} + \lambda\mathbf{I})^{-1} \approx (\mathbf{Q}_{\mathbf{A}_{l-1}} \otimes \mathbf{Q}_{\mathbf{S}_l})(\mathbf{\Lambda}_{l,\text{KFAC}} + \lambda\mathbf{I})^{-1}(\mathbf{Q}_{\mathbf{A}_{l-1}} \otimes \mathbf{Q}_{\mathbf{S}_l})^\top$ where $\lambda$ is a damping hyperparameter. The EKFAC approximation $\mathbf{F}_{\text{EKFAC}}$ is an improvement over KFAC using the diagonal matrix $\mathbf{\Lambda}_{l,\text{EKFAC}}$ instead, whose $i$th entry along the diagonal is:

$$\mathbf{\Lambda}_{l,\text{EKFAC},i} = \mathbb{E}\big[\big((\mathbf{Q}_{\mathbf{A}_{l-1}} \otimes \mathbf{Q}_{\mathbf{S}_l})^\top \mathcal{D}\boldsymbol{\theta}_l\big)_i^2\big], \tag{4}$$

where $\mathcal{D}\boldsymbol{\theta}_l$ are the pseudogradients with respect to the parameters in layer $l$.

To illustrate the compute and memory savings, for a $P$-layer multi-layer perceptron with $\tilde{D}$ input dimensions and $\tilde{D}$ output dimensions for all layers, due to the block-diagonal approximation, the EKFAC eigendecomposition only costs $O(P\tilde{D}^3) = O(D^{\frac{3}{2}})$ time and storing its statistics requires $O(P\tilde{D}^2) = O(D)$ memory [49].[7] Although EKFAC significantly reduces the time and space complexity of the eigendecomposition, it results in a biased iHVP. See Appendix B for a more detailed discussion of EKFAC.

**Computing the iHVP with LiSSA**  In contrast to EKFAC, iterative methods such as LiSSA are based on Neumann series iterations (NI) [24]. NI do not require EKFAC's assumptions, and thus in principle can produce exact iHVPs as more compute is applied. For an invertible matrix $\mathbf{A} \in \mathbb{R}^{D \times D}$ with $\|\mathbf{I} - \mathbf{A}\|_2 < 1$, the Neumann series is approximated as $\mathbf{A}^{-1} = \sum_{j=0}^\infty (\mathbf{I} - \mathbf{A})^j$, which is a generalization of the geometric series $a^{-1} = \sum_{j=0}^\infty (1 - a)^j$ for $|1 - a| < 1$. By substituting $\mathbf{A}$ with a scaled positive-definite damped Generalized Gauss-Newton Hessian (GGN) $\alpha(\mathbf{G} + \lambda\mathbf{I})$, and multiplying both sides by any $\mathbf{v} \in \mathbb{R}^D$, we obtain: $\frac{1}{\alpha}(\mathbf{G} + \lambda\mathbf{I})^{-1}\mathbf{v} = \sum_{j=0}^\infty (\mathbf{I} - \alpha\mathbf{G} - \alpha\lambda\mathbf{I})^j \mathbf{v}$. Here $\lambda$ is a positive scalar known as the damping term and $\alpha > 0$ is the learning rate hyperparameter. The learning rate must satisfy $\alpha < \frac{1}{\sigma_{\max}(\mathbf{G}) + \lambda}$, where $\sigma_{\max}(\mathbf{G})$ is the largest eigenvalue of $\mathbf{G}$ so that $\|\mathbf{I} - \mathbf{A}\|_2 < 1$. We can then approximate the iHVP $\frac{1}{\alpha}(\mathbf{G} + \lambda\mathbf{I})^{-1}\mathbf{v}$ via the iterative update:

$$\mathbf{v}_{k+1} \leftarrow \mathbf{v}_k - \alpha(\mathbf{G} + \lambda\mathbf{I})\mathbf{v}_k + \mathbf{v}, \tag{5}$$

which satisfies the property $\mathbf{v}_k \to \frac{1}{\alpha}(\mathbf{G} + \lambda\mathbf{I})^{-1}\mathbf{v}$ as $k \to \infty$. Computing $\mathbf{G}$ requires two backward passes over the whole dataset,[8] so an unbiased estimate $\tilde{\mathbf{G}}_k$ of $\mathbf{G}$ is usually used instead by sampling a mini-batch with replacement, which we refer to as *stochastic Neumann series iterations* (SNI). Compared to other iterative methods like conjugate gradient (CG) [51], SNI is typically preferred to compute the iHVP for TDA since CG tends to struggle with stochastic gradients [17, 26, 52]. LiSSA reduces the variance in SNI by taking an average over multiple trials.

## 2.2 Training Data Attribution with Influence Functions

Influence functions [16, 53] are derived under the assumption that $\mathcal{J}$ is strictly convex in $\boldsymbol{\theta}$ and twice differentiable. Let $\boldsymbol{\theta}^\star := \arg\min_{\boldsymbol{\theta}} \mathcal{J}(\boldsymbol{\theta}, \mathcal{D})$ be the optimal parameters over $\mathcal{D}$. We can define the objective after downweighting a training example $\boldsymbol{z}_m$ by $\epsilon$ as: $\mathcal{Q}(\boldsymbol{\theta}, \epsilon) := \mathcal{J}(\boldsymbol{\theta}, \mathcal{D}) - \frac{\epsilon}{N}\mathcal{L}(\boldsymbol{\theta}, \boldsymbol{z}_m)$ where $\epsilon$ is a scalar that specifies the amount of downweighting. When $\epsilon = 0$, this corresponds to the original objective $\mathcal{J}$. We can then define the optimal parameters *after* downweighting as a function of $\epsilon$: $r(\epsilon) = \arg\min_{\boldsymbol{\theta} \in \mathbb{R}^D} \mathcal{Q}(\boldsymbol{\theta}, \epsilon)$. When $\boldsymbol{z}_m \in \mathcal{D}$ and $\epsilon = 1$, this corresponds to the downweighted objective in which $\boldsymbol{z}_m$ is removed from $\mathcal{D}$. Typically, $\epsilon$ is assumed to be small, and we can approximate the leave-one-out (LOO) parameter change as $\boldsymbol{\theta}^\star(\mathcal{D} \setminus \{\boldsymbol{z}_m\}) - \boldsymbol{\theta}^\star(\mathcal{D}) \approx \frac{\mathrm{d}r}{\mathrm{d}\epsilon}\big|_{\epsilon=0}$, where $\frac{\mathrm{d}r}{\mathrm{d}\epsilon}\big|_{\epsilon=0} = \frac{1}{N}\mathbf{H}^{-1}\nabla_{\boldsymbol{\theta}}\mathcal{L}(\boldsymbol{\theta}^\star, \boldsymbol{z}_m)$ and $\mathbf{H} := \nabla_{\boldsymbol{\theta}}^2 \mathcal{J}(\boldsymbol{\theta}^\star, \mathcal{D})$ denotes the Hessian of the training objective at the optimal parameters. To approximate the effect on the measurement function $f_{\boldsymbol{z}_q}$, we invoke the chain rule: $f_{\boldsymbol{z}_q}(\boldsymbol{\theta}^\star(\mathcal{D} \setminus \{\boldsymbol{z}_m\})) - f_{\boldsymbol{z}_q}(\boldsymbol{\theta}^\star(\mathcal{D})) \approx \frac{1}{N}\nabla_{\boldsymbol{\theta}} f_{\boldsymbol{z}_q}(\boldsymbol{\theta}^\star)^\top \mathbf{H}^{-1}\nabla_{\boldsymbol{\theta}}\mathcal{L}(\boldsymbol{\theta}^\star, \boldsymbol{z}_m)$, which also gives the first-order Taylor approximation of $f_{\boldsymbol{z}_q}(\boldsymbol{\theta}^\star(\mathcal{D} \setminus \{\boldsymbol{z}_m\}))$ after rearranging terms. When applying this approximation to neural networks in which the convexity assumption does not

---

[7]The time and space complexity of ordinary eigendecomposition is $O(D^3)$ and $O(D^2)$ respectively, and $P\tilde{D}^3$ typically is much smaller than $D^3$. This analysis treats $P$ as constant.

[8]An efficient implementation with $O(D)$ time and space complexity requires a Jacobian-vector product and a vector-Jacobian product [50].

hold, $\mathbf{H}$ may not be invertible, so $\mathbf{H}$ is typically approximated with the damped GGN $\mathbf{G} + \lambda\mathbf{I}$ [54], which is always positive definite for $\lambda > 0$, and tends to work well in practice [7, 11, 19, 55].[9] With this substitution, the influence functions *attribution score* is:

$$\tau_{\mathrm{IF}}(\boldsymbol{z}_m, \boldsymbol{z}_q, \mathcal{D}) \coloneqq \nabla_{\boldsymbol{\theta}} f_{\boldsymbol{z}_q}(\boldsymbol{\theta}^\star)^\top (\mathbf{G} + \lambda\mathbf{I})^{-1} \nabla_{\boldsymbol{\theta}} \mathcal{L}(\boldsymbol{\theta}^\star, \boldsymbol{z}_m). \tag{6}$$

When applying influence functions to neural networks, the model may not have fully converged, so we typically compute the gradients and the GGN in Equation 6 with the final parameters $\boldsymbol{\theta}^s$ instead of $\boldsymbol{\theta}^\star$. We can also ensemble influence functions for better TDA performance, by training models with various seeds $\xi$ and averaging over $\tau_{\mathrm{IF}}$ for each seed to get an ensembled score [19, 22] (details in Appendix B). Ensembling for other TDA methods, such as unrolled differentiation, can be done analogously.

The iHVP in Equation 6 was originally computed with LiSSA by Koh and Liang [17] and has the drawback that its iterative procedure must be carried out once for each vector in the iHVP. Fortunately, it is frequently the case that $|\mathcal{D}_{\mathrm{query}}| \ll |\mathcal{D}|$ [7, 36, 56], so by choosing $\mathbf{v} \coloneqq \nabla_{\boldsymbol{\theta}} f_{\boldsymbol{z}_q}(\boldsymbol{\theta}^s)$ and first computing $\nabla_{\boldsymbol{\theta}} f_{\boldsymbol{z}_q}(\boldsymbol{\theta}^s)^\top (\mathbf{G} + \lambda\mathbf{I})^{-1}$ in Equation 6,[10] we can reduce the number of iterative procedures to $|\mathcal{D}_{\mathrm{query}}|$ for the influence function computation. Plugging these values into Equation 5, we can compute $\nabla_{\boldsymbol{\theta}} f_{\boldsymbol{z}_q}(\boldsymbol{\theta}^s)^\top (\mathbf{G} + \lambda\mathbf{I})^{-1}$ via the iterative update:[11]

$$\boldsymbol{\theta}_{k+1} \leftarrow \boldsymbol{\theta}_k - \alpha(\tilde{\mathbf{G}}_k + \lambda\mathbf{I})\boldsymbol{\theta}_k + \alpha\nabla_{\boldsymbol{\theta}} f_{\mathbf{z}_q}(\boldsymbol{\theta}^s). \tag{7}$$

While $|\mathcal{D}_{\mathrm{query}}|$ sounds like a large number of iterative procedures, in practice no $\mathcal{D}_{\mathrm{query}}$ exists, and instead queries are run when the user wants to understand specific behavior pertaining to $\boldsymbol{z}_q$. Nevertheless, to get a good approximation of $\nabla_{\boldsymbol{\theta}} f_{\boldsymbol{z}_q}(\boldsymbol{\theta}^s)^\top (\mathbf{G} + \lambda\mathbf{I})^{-1}$ for any particular $\boldsymbol{z}_q$ may require *thousands* of iterations [7, 17], limiting its scalability. Furthermore, tuning the hyperparameter for SNI is difficult [17, 32, 33] due to the ill-conditioning and stochasticity of the gradients. Once the iHVP is computed, its dot product with $\nabla_{\boldsymbol{\theta}} \mathcal{L}(\boldsymbol{\theta}^s, \boldsymbol{z}_m)$ for every $\boldsymbol{z}_m$ in consideration is taken. If the goal is to simply compute the influence of $\boldsymbol{z}_m$ on $\boldsymbol{z}_q$ for *given* $\boldsymbol{z}_m$ and $\boldsymbol{z}_q$, then the cost of the dot product is minimal. However, if the goal is to search for the most influential points in $\boldsymbol{z}_m \in \mathcal{D}$ on $\boldsymbol{z}_q$, then we must take the dot product with *every* training example gradient in $\mathcal{D}$, which amounts to a backward pass over the entire training dataset and can be a substantial component of the cost of computing influence functions.[12] Arguably, the latter goal is more prevalent in settings such as data-centric model debugging and interpretability, where insight into model behavior is given by retrieving highly influential training samples [57–61].

## 3 Introducing EKFAC-Accelerated Neumann Series Iterations for TDA

We now have laid sufficient groundwork to introduce a novel algorithm called **ASTRA**, which preconditions SNI updates with EKFAC. In this section, we present the ASTRA update rule, discuss its computational costs, and discuss extensions to unrolled-differentiation-based TDA.

**Preconditioning Stochastic Neumann Series Iterations with EKFAC** The SNI update rule in Equation 7 can be viewed as performing mini-batch gradient descent on the quadratic objective [23, 62]:

$$h_{f_{\boldsymbol{z}_q}}(\boldsymbol{\theta}) \coloneqq \frac{1}{2}\boldsymbol{\theta}^\top (\mathbf{G} + \lambda\mathbf{I})\boldsymbol{\theta} - \boldsymbol{\theta}^\top \nabla_{\boldsymbol{\theta}} f_{\boldsymbol{z}_q}(\boldsymbol{\theta}^s). \tag{8}$$

It is well-known that for a converged neural network, the curvature matrix in the objective in Equation 8 is typically ill-conditioned [30, 31], which presents challenges for iterative methods. To improve the conditioning of $h_{f_{\boldsymbol{z}_q}}(\boldsymbol{\theta})$, we introduce an algorithm which computes accurate iHVPs for use in TDA called EKFAC-**A**ccelerated Neumann **S**eries Iterations for **TR**aining Data **A**ttribution (ASTRA) by applying preconditioning. The resulting update rule is:

$$\boldsymbol{\theta}_{k+1} \leftarrow \boldsymbol{\theta}_k - \alpha(\mathbf{P} + \tilde{\lambda}\mathbf{I})^{-1}(\tilde{\mathbf{G}}_k + \lambda\mathbf{I})\boldsymbol{\theta}_k + \alpha(\mathbf{P} + \tilde{\lambda}\mathbf{I})^{-1}\nabla_{\boldsymbol{\theta}} f_{\boldsymbol{z}_q}(\boldsymbol{\theta}^s). \tag{9}$$

---

[9]We will use the approximation $\mathbf{G} \approx \mathbf{H}$ throughout, and our use of the term iHVP will generally refer to both the inverse Hessian-vector product and the inverse Gauss-Newton-Hessian-vector product.

[10]This uses the fact that $\mathbf{G}$ is symmetric. $(\nabla_{\boldsymbol{\theta}} f_{\boldsymbol{z}_q}(\boldsymbol{\theta}^s)^\top (\mathbf{G} + \lambda\mathbf{I})^{-1})^\top = (\mathbf{G} + \lambda\mathbf{I})^{-1}\nabla_{\boldsymbol{\theta}} f_{\boldsymbol{z}_q}(\boldsymbol{\theta}^s)$.

[11]We emphasize that $\tilde{\mathbf{G}}_k$ and $\nabla_{\boldsymbol{\theta}} f_{\boldsymbol{z}_q}(\boldsymbol{\theta}^s)$ are computed using the final parameters $\boldsymbol{\theta}^s$. We use $\boldsymbol{\theta}_k$ to denote the iterates for SNI since it has the same dimensions as the model's parameters.

[12]Procedures such as TF-IDF filtering exist to prune the potentially vast training dataset [7].

While a number of choices of preconditioners exist, we choose the FIM computed with EKFAC (i.e., $\mathbf{P} := \mathbf{F}_{\text{EKFAC}}$) for the following reasons: First, the computation cost of the EKFAC eigendecomposition is usually much cheaper than full matrix inversion, and its eigendecomposition statistics can be stored compactly and shared across all $|\mathcal{D}_{\text{query}}|$ optimization problems, since the value of $\mathbf{G}$ is the same across all objectives. Second, this choice has a close connection with EKFAC influence functions: Equation 5 suggests initializing $\boldsymbol{\theta}_0$ as $\nabla_{\boldsymbol{\theta}} f_{\boldsymbol{z}_q}(\boldsymbol{\theta}^s)$, which is frequently done in public implementations [17]. Observe that if we initialize $\boldsymbol{\theta}_0 \leftarrow \mathbf{0}$, choose $\tilde{\lambda} = \lambda$ and a learning rate $\alpha = 1$, we arrive at the iHVP which would be approximated by EKFAC after one step of ASTRA, resulting in the same TDA prediction as EKFAC-IF.[13] We hypothesize that further training using the update rule in Equation 9 will improve the iHVP approximation, a claim we validate empirically in Section 5. This update rule assumes using mini-batch gradient descent, but our formulation is compatible with other optimization algorithms as well.

**Time Complexity of ASTRA**   Computing the update in Equation 9 involves explicitly constructing neither $(\mathbf{P} + \tilde{\lambda}\mathbf{I})^{-1}$ nor $(\mathbf{G} + \lambda\mathbf{I})$. Instead, we use Hessian-vector products [50] and first compute $(\tilde{\mathbf{G}}_k + \lambda\mathbf{I})\boldsymbol{\theta}_k$. We can then compute $(\mathbf{P} + \tilde{\lambda}\mathbf{I})^{-1}(\tilde{\mathbf{G}}_k + \lambda\mathbf{I})\boldsymbol{\theta}_k$ using the EKFAC preconditioner $\mathbf{P} := \mathbf{F}_{\text{EKFAC}}$, so the *incremental* time complexity of *each iteration* in our algorithm is $O(\mathcal{B}D)$ where $\mathcal{B}$ is the mini-batch size used to sample $\tilde{\mathbf{G}}_k$. For both EKFAC-IF and ASTRA-IF, computing $\mathbf{F}_{\text{EKFAC}}$ is necessary to compute the iHVP, which only needs to be done once per model and can be shared among all queries. When we need to search the entire dataset for highly influential training examples, both methods also need to compute the dot product of the resulting iHVP with the training example gradients over all $\boldsymbol{z}_m$ in consideration, as discussed in Section 2.2. In our experiments, ASTRA-IF required only a few hundred incremental iterations, so its iterative component is a relatively small additional cost per query compared to the total cost to compute EKFAC-IF.

**Application to Unrolled Differentiation**   In addition to influence functions, we also study the use of ASTRA in the context of an unrolling-based TDA method. Influence functions make assumptions such as model convergence and unique optimal parameters in its derivation, which may not be satisfied in practice [17, 36]. In contrast, unrolled differentiation methods such as SOURCE sidestep this limitation by differentiating through the training trajectory. Here, we only sketch our approach to apply ASTRA to SOURCE, deferring the full discussion of the SOURCE derivation to Appendix B and the details of ASTRA-SOURCE to Appendix C. SOURCE approximates differentiating through the training trajectory by partitioning it into $L$ segments, assuming stationary and independent GGNs and gradients within each. Its approximation of the first order effect of downweighting a training example contains $L$ different finite series involving the GGN, each of which can be approximated with an iHVP. Similarly to ASTRA-IF, ASTRA-SOURCE improves this approximation by repurposing the EKFAC decompositions as preconditioners, which would have been needed to be computed to implement SOURCE anyways.

# 4   Relationship to Existing Works

**TDA Methods using iHVPs**   Both influence functions and unrolled differentiation can be viewed as belonging to a family of gradient-based TDA methods (for a survey see [58]), which both have a connection with iHVPs. Many gradient-based TDA methods are variants of the influence functions method proposed by Koh and Liang [17] with aims to improve its computational cost [7, 22, 33], by using techniques such as EKFAC for iHVP approximation [7], Arnoldi iterations [33], gradient projection [1, 22], and rank-one updates [63], instead of iterative algorithms such as LiSSA [23] or CG [51, 64], since the former is expensive and hard to tune [17, 32, 33], and the latter struggles with stochastic gradients [65]. Unrolled differentiation addresses the key derivation assumptions underlying influence functions – namely, the convexity of the training objective and the convergence of the final model parameters [36]. Methods include SGD-influence [4], HYDRA [18], SOURCE [19], DVEmb [20] and MAGIC [21], which all differentiate through the training trajectory, and only differ in their approximations. Rather than comparing the TDA algorithms themselves, our goal in this paper is to show that substantial TDA performance improvements can be achieved by using better iHVP approximations. In some public comparisons, LiSSA-based influence functions perform poorly

---

[13]Going forward, we therefore directly initialize ASTRA with $\boldsymbol{\theta}_0 \leftarrow (\mathbf{P} + \tilde{\lambda}\mathbf{I})^{-1}\nabla_{\boldsymbol{\theta}} f_{\boldsymbol{z}_q}(\boldsymbol{\theta}^s)$.

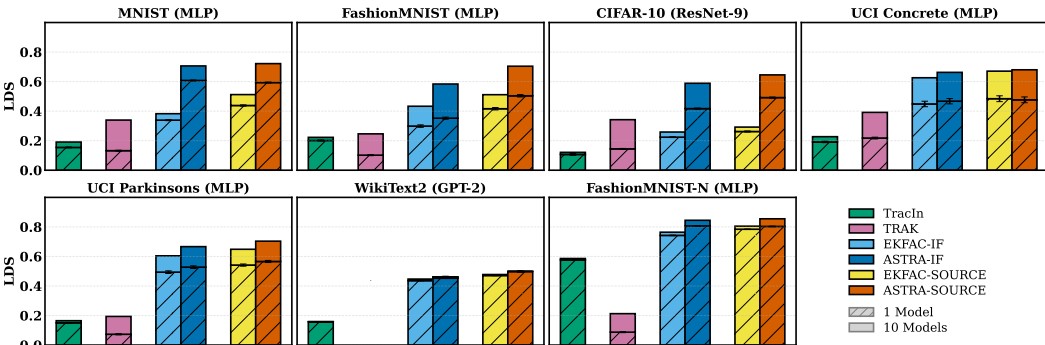

Figure 2: **TDA Performance.** Single model ASTRA-IF and ASTRA-SOURCE beat EKFAC-based counterparts in most settings, as well as other TDA methods such as TracIn [83] and TRAK [22] when measured by average LDS over the query set $\mathcal{D}_{\text{query}}$. ASTRA also enjoys a larger performance boost from ensembling. Improvement is particularly large for convolution architectures such as ResNet-9. Error bars (where available) indicate 1 standard error. We omit TRAK for GPT-2 due to lack of public implementations.

[22, 63, 66], sometimes even worse than dot products[14] [63, 67] and many have opted to instead use EKFAC-IF as their method or baseline of choice [1, 11, 19]. In principle, EKFAC-IF is only an *approximation* of what LiSSA-based influence functions attempts to compute, since the latter is only constrained by solver error while the former makes assumptions on the structure of the curvature matrix. We show in this paper that indeed, accurately solving the iHVP typically produces better TDA performance than the EKFAC solution, which is what our algorithm ASTRA addresses.

**iHVPs beyond TDA** iHVPs can also be found in higher-order optimization algorithms such as Newton's method [28], quasi-Newton methods [68–71], natural gradient descent [65, 72, 73], KFAC [26], and Hessian-free optimization [65, 74], which computes Hessian-vector products iteratively with CG [51, 64]. Influence functions can also be cast as a bilevel optimization problem [25, 29, 44, 75], which can be solved via implicit differentiation or unrolled differentiation. Since iHVPs show up frequently in machine learning, there is motivation to adapt and develop iHVP computation techniques such as SNI and EKFAC. While these two methods are well-established and preconditioning is a well-established technique in optimization [26, 72], to the best of our knowledge, no prior TDA method has combined these methods to compute the iHVP. For extended related works, see Appendix D.

## 5 Performance Comparisons

This section aims to answer the following questions: 1) Do ASTRA-IF and ASTRA-SOURCE outperform their EKFAC-based counterparts? 2) Is ASTRA substantially faster than vanilla SNI? To answer these questions, we run experiments in a number of settings. For regression tasks, we use the UCI datasets Concrete and Parkinsons [76] trained with a multi-layer perceptron (MLP). For classification tasks, we use CIFAR-10 [77] trained with ResNet-9 [78], MNIST [79] and FashionMNIST[80] trained with MLPs, and GPT-2 [81] fine-tuned with WikiText-2 [82]. We also include a non-converged setting, FashionMNIST-N, introduced by Bae et al. [19], for which SOURCE was specifically designed; in this setting, 30% of the training examples were randomly labeled, and the model was trained for only three epochs to avoid overfitting [19]. In addition to comparing against EKFAC-IF and EKFAC-SOURCE, we also compare against two popular TDA methods TracIn [83] and TRAK [22]. Details for all experiments can be found in Appendix F.

**Evaluating Training Data Attribution Performance** We evaluate the performance of our TDA algorithms on a popular evaluation metric called Linear Datamodeling Score (LDS) [22], and use mean absolute error as the measurement function for regression tasks and correct-class margin for classification tasks in line with past works [19, 22]. LDS measures a TDA algorithm's ability to predict the outcome of counterfactual retraining on a subset of data. Given a collection of uniformly

---

[14]This TDA method, sometimes called Hessian-free [63, 67], simply takes the dot product between the training gradient and the query gradient and is equivalent to influence functions if the damped GGN is set to the identity.

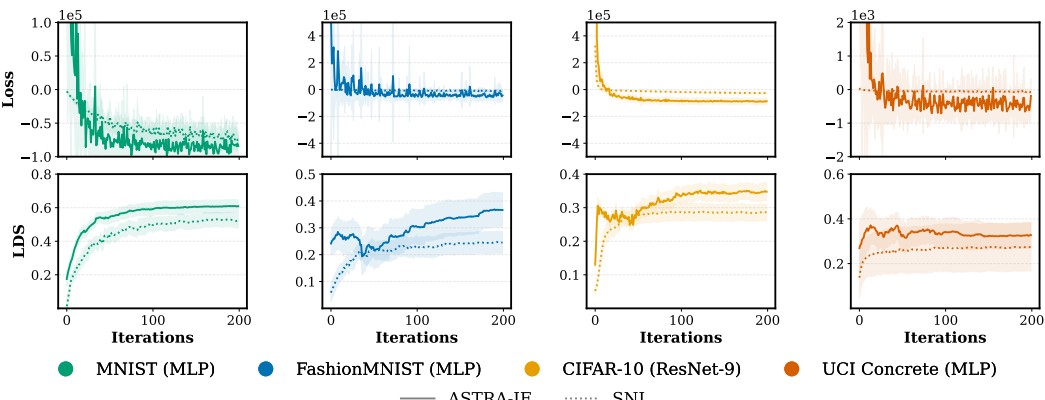

Figure 3: **Training Curves**. Loss and LDS curves for SNI and ASTRA measured over 10 seeds (shaded region = 1 standard error) on an arbitrary query point $z_q$, using influence functions as the TDA method. SNI makes slower progress compared to ASTRA as measured by LDS.

randomly sampled subsets $\{S_1, \ldots, S_M : S_i \subset \mathcal{D}\}$ of a fixed size, typically a fraction $\beta$ of $\mathcal{D}$ and a measurement function $f_{z_q}$, the LDS scores a TDA method $\tau$ as follows:

$$\text{LDS}(\tau, z_q) := \rho\big(\underbrace{\big[\mathbb{E}_\xi[f_{z_q}(\theta^s(S_j; \xi))] : j \in [M]\big]}_{\substack{\textit{expected} \text{ ground-truth predictions on } z_q \\ \text{using models trained on various } S_j}}, \underbrace{\big[\Gamma_\tau(S_j, z_q; \mathcal{D}) : j \in [M]\big]}_{\substack{\text{TDA method's predictions on } z_q \\ \text{for various } S_j}}\big), \qquad (10)$$

where $\rho$ denotes the Spearman correlation [37], and the group influence $\Gamma_\tau(S_j, z_q; \mathcal{D})$ is defined linearly as: $\Gamma_\tau(S_j, z_q; \mathcal{D}) = \sum_{z_i \in S_j} \tau(z_i, z_q, \mathcal{D})$. To compute the ground-truth to which we compare our TDA method, we need to retrain a model many times both over various subsets $S_i$, and also over random seeds $\xi$ for every subset to obtain a good estimate of the expectation in Equation 10, which can be quite noisy [19, 43]. For example, for the experiment involving GPT-2, computing ground-truth involved fine-tuning 1000 models. We report the *average* LDS $\frac{1}{|\mathcal{D}_{\text{query}}|}\sum_{z_q \in \mathcal{D}_{\text{query}}} \text{LDS}(\tau, z_q)$ over a test set $\mathcal{D}_{\text{query}}$ containing 100 query points. We randomly sample $M = 100$ subsets with a subsampling fraction of $\beta = 0.5$ in line with previous works [1, 19]. We discuss other evaluation methods in Appendix E.

**LDS evaluation of ASTRA**   Figure 2 compares LDS across various TDA methods. We compare ASTRA-IF and ASTRA-SOURCE with their respective EKFAC-versions. For each setting, EKFAC-IF and ASTRA-IF use the same damping value implied by SOURCE for comparability (details in Appendix F). In almost all settings, ASTRA improves TDA performance as measured by LDS, strongly suggesting that better iHVP approximations are responsible. We also observe an especially large improvement over EKFAC for CIFAR-10 trained on ResNet-9 [78]. When applied to convolution layers, EKFAC makes additional simplifying assumptions,[15] which can cause EKFAC-IF and EKFAC-SOURCE to underperform on architectures involving convolution layers – an issue that ASTRA effectively addresses. Figure 2 also reveals increased benefits from ensembling when applying ASTRA, which computes an unbiased estimator of the iHVP. In some cases, such as FashionMNIST, the advantages of using precise iHVPs become much more pronounced in conjunction with ensembling. We hypothesize that this is due to the various bottlenecks in TDA methods: in some cases, the primary performance bottleneck lies in computing the iHVP accurately; in others, it stems from other factors such as the method's underlying assumptions (e.g., unique optimal parameters), which can be mitigated by ensembling.

**ASTRA speeds up iHVP approximation**   The top row of Figure 3 shows the loss curves for SNI and ASTRA as each iHVP solver progresses. The bottom row corresponds to the LDS that influence functions achieves based on the progress of each iHVP solver. For SNI, we follow public implementations [17], which typically initialize SNI using the query gradient $\nabla_\theta f_{z_q}(\theta^s)$, and results

---

[15]In addition to layer-wise independence and independence of activations and pseudogradients [26], it assumes spatially uncorrelated derivatives and spatial homogeneity [84].

in an initial influence functions prediction that approximates the damped-GGN with the identity in Equation 6. For both methods, we conduct a hyperparameter sweep for the learning rate over $10^0$, $\ldots$, $10^{-5}$ in steps of one order of magnitude, and use the best hyperparameter based on the average training loss performance on the same query point over the last 10 iterations. We find that EKFAC preconditioning reduces the notorious challenge of tuning learning rates [17, 32, 33] – in all of the settings reported in Figure 3 the learning rate for ASTRA used was $10^{-2}$. In comparison, the reported (and best) SNI learning rates were $1, 0.1, 0.01, 0.1$ for MNIST, FashionMNIST, CIFAR-10, and UCI Concrete respectively and LDS performance was very sensitive to the learning rate. Figure 3 shows that SNI makes slow progress while ASTRA usually converges in fewer than 200 iterations.

## 6 Investigating the Role of Low Curvature Directions in Influence Functions Performance

Our results in the previous section lead us to hypothesize that preconditioning accelerates convergence in directions of low curvature, which is important for influence function performance. We can analyze how directions of low curvature are affected by NI (without preconditioning) when truncated early, something that is tempting to do as it usually takes long to converge. We derive the following expression for the truncated Neumann series with $J$ iterations:

$$\alpha \sum_{j=0}^{J-1} (\mathbf{I} - \alpha\mathbf{G} - \alpha\lambda\mathbf{I})^j = (\mathbf{G} + \lambda\mathbf{I})^{-1}(\mathbf{I} - (\mathbf{I} - \alpha\mathbf{G} - \alpha\lambda\mathbf{I})^J) \tag{11}$$

$$\approx (\mathbf{G} + (\lambda + \alpha^{-1}J^{-1})\mathbf{I})^{-1}, \tag{12}$$

where the equality utilizes the definition for finite series, and the approximation is identical to that used in [19] (see Appendix G for the derivation). From the last equation, we can see that truncating Neumann series effectively adds an *implicit* damping term of $1/\alpha J$, which disproportionately affects directions of low curvature.

This insight prompts an investigation into the role of low curvature directions in influence functions performance since, in addition to the implicit damping effect mentioned above, some methods may discard them when projecting gradients into lower-dimensional subspaces [1, 33]. Let $\mathbf{F}_{\text{EKFAC}} = \hat{\mathbf{Q}}\hat{\mathbf{D}}\hat{\mathbf{Q}}^\top$ be the EKFAC eigendecomposition of the GGN at the final parameters. We study the importance of directions of varying curvature by doing the following: We bin the $D$ eigenvalues given by $\mathbf{F}_{\text{EKFAC}}$ into 5 bins labeled $S_1, S_2, \ldots, S_5$, where each bin $S_i$ holds all eigenvalues larger than $10^{-i}$. Let $\hat{\mathbf{Q}}_{S_i} \in \mathbf{R}^{D \times |S_i|}$ be the projection matrix whose columns are the associated orthonormal eigenvectors of the eigenvalues in each bin. We investigate the values of $h_{f_{\boldsymbol{z}_q}}^{S_i}(\boldsymbol{\theta}_k)$ and the corresponding LDS for each $S_i$ where $h_{f_{\boldsymbol{z}_q}}^{S_i}$ is defined as:

$$h_{f_{\boldsymbol{z}_q}}^{S_i}(\boldsymbol{\theta}) := \frac{1}{2}(\hat{\mathbf{Q}}_{S_i}^\top \boldsymbol{\theta})^\top \hat{\mathbf{Q}}_{S_i}^\top (\mathbf{G} + \lambda\mathbf{I})\hat{\mathbf{Q}}_{S_i}(\hat{\mathbf{Q}}_{S_i}^\top \boldsymbol{\theta}) - (\hat{\mathbf{Q}}_{S_i}^\top \boldsymbol{\theta})^\top \hat{\mathbf{Q}}_{S_i}^\top \nabla_{\boldsymbol{\theta}} f_{\boldsymbol{z}_q}(\boldsymbol{\theta}^s). \tag{13}$$

We use the EKFAC eigendecomposition since the true eigendecomposition is intractable for the settings we report. We conduct the experiment in the MNIST and FashionMNIST settings and use a small damping hyperparameter of $\lambda = 10^{-4}$ to be able to observe the impact of directions of low curvature on influence functions performance (details in Appendix F).

The top row of Figure 4 shows the outcome when we run ASTRA and SNI on the objective in Equation 8 to obtain a sequence of $\boldsymbol{\theta}_k$, and plot the value of each $h_{f_{\boldsymbol{z}_q}}^{S_i}(\boldsymbol{\theta}_k)$ for each $S_i$. The bottom row shows the LDS, which are computed by projecting the iterates $\boldsymbol{\theta}_k$ into each curvature subspace defined by $S_i$ and computing influence functions using these projected vectors, which we can write as:

$$\tau_{\text{PROJ-IF},k}(\boldsymbol{z}_m, \boldsymbol{z}_q, \mathcal{D}) := (\hat{\mathbf{Q}}_{S_i}^\top \boldsymbol{\theta}_k)^\top \hat{\mathbf{Q}}_{S_i}^\top \nabla_{\boldsymbol{\theta}} \mathcal{L}(\boldsymbol{\theta}^s, \boldsymbol{z}_m). \tag{14}$$

Figure 4 reveals that low curvature directions play a large role in the performance of influence functions: projecting to high-curvature eigenspaces degrades LDS performance, as evidenced by the large vertical gaps between lines representing different level of curvature. When the iHVP is computed via SNI, large eigenvalue directions converge quickly during training, but it takes longer for small eigenvalue directions to converge, as evidenced by the earlier plateau of the loss curves in

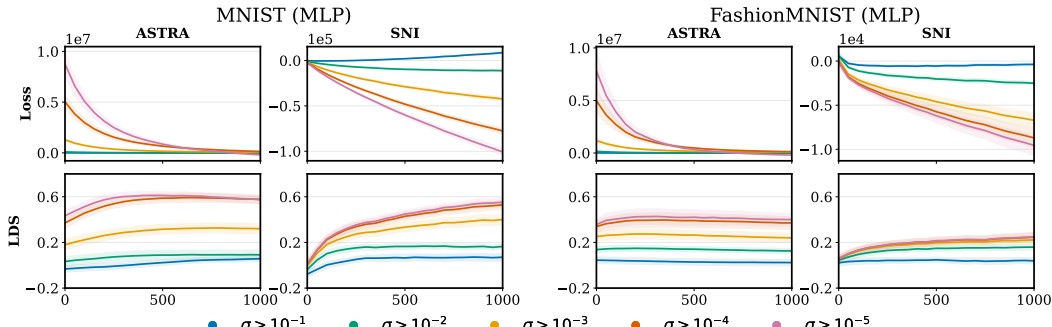

Figure 4: **Training Curves in Various Eigenspaces**. **Top:** The values $h_{f z_q}^{S_i}$ as ASTRA and SNI train on the objective in Equation 8 for an arbitrary $z_q$. The subspaces represented by $S_1, \ldots, S_5$ are spanned by eigenvectors with eigenvalues $\sigma > 10^{-1}, \ldots, \sigma > 10^{-5}$ respectively. The loss of subspaces with large eigenvalue directions tend to plateau first, followed by subspaces with smaller eigenvalue directions, especially for SNI, which does not use preconditioning. **Bottom:** LDS of influence functions after projecting to corresponding subspace. The objective for *high* curvature directions plateaus first; continued training further decreases the objective in progressively lower curvature directions, yielding LDS gains even after high curvature directions have converged. Shaded region = 1 standard error.

the high-curvature subspaces in the top row of Figure 4. Nevertheless, as the solution progresses in low-curvature directions, LDS rises substantially, evidenced by the growing gap between lines representing large and small levels of curvature in the bottom row of Figure 4. While the slower convergence in low-curvature directions is present for ASTRA, it is substantially diminished due to the preconditioning. Our results highlight that the behavior of estimators in low-curvature subspaces may be a substantial factor in the performance of TDA methods.[16]

# 7    Conclusion

We presented an algorithm ASTRA that combines the EKFAC preconditioner with SNI for TDA. We compared ASTRA-IF and ASTRA-SOURCE with their EKFAC-based counterparts in a variety of settings. In many settings, TDA performance measured by LDS improved substantially, especially for convolution architectures. We find that in general, a more accurate iHVP approximation increases the efficacy of ensembling. ASTRA is easier to tune and converges faster than SNI. Compared with EKFAC, it only incrementally costs hundreds of iterations in our experiments as it leverages the same eigendecomposition. We conclude this paper by providing insights into how various curvature directions affect influence functions performance. We show that low curvature directions are important for good influence functions performance by using the EKFAC decomposition to analyze the quadratic objective, a technique that may be of independent interest outside of TDA. Overall, the technical contributions of this work should lead to improved performance in real-world problems such as data curation and interpreting model behavior, among other applications. We discuss limitations and broader implications of our work in Appendix H.

## Acknowledgements

We gratefully acknowledge funding from the Natural Sciences and Engineering Research Council of Canada (NSERC) and the Canada CIFAR AI Chairs Program. Resources used in preparing this research were provided, in part, by the Province of Ontario, the Government of Canada through CIFAR, and companies sponsoring the Vector Institute for Artificial Intelligence (https://vectorinstitute.ai/partnerships/). We thank the Schwartz Reisman Institute for Technology and Society for providing a rich multi-disciplinary research environment. RG and SM acknowledge support from the Canada CIFAR AI Chairs program and from the Natural Sciences

---

[16]We note that the difference in performance between ASTRA and SNI depends on the damping hyperparameter, and since the damping in this experiment is smaller, the performance boost from ASTRA compared to SNI is notably larger than in Figure 3.

and Engineering Research Council of Canada (NSERC). RG acknowledges support from Open Philanthropy and the Schmidt Sciences AI2050 Fellows Program.

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

# A  Notation & Acronyms

## A.1  Notation

| Notation | Description |
|---|---|
| $B$ | Batch size in mini-batch gradient descent |
| $\mathcal{B}$ | Batch size for stochastic Neumann series iterations |
| $D$ | Number of parameters in neural network |
| $J$ | Number of iterations for SNI |
| $M$ | Number of subsets (masks) to sample for LDS computation |
| $N$ | Number of training data points, $N = |\mathcal{D}|$ |
| $P$ | Number of layers in a neural network. |
| $R$ | Number of trials (repeats) for LiSSA. |
| $\mathcal{D} = \{z_i\}_{i=1}^N$ | Training dataset |
| $\mathcal{D}_{\text{query}}$ | Query dataset, used to benchmark TDA algorithms and small in practice |
| $\mathcal{S}$ | Data subset of the training dataset |
| $z_i$ | An arbitrary $i$-th training example |
| $z_m \in \mathcal{D}$ | A training example from the dataset $\mathcal{D}$ |
| $z_q$ | A query data point |
| $\mathbf{x}^{(i)}$ | The inputs (feature vector) of the $i$-th training example |
| $\mathbf{z}^{(i)}$ | The neural network output for the $i$-th training example |
| $\mathbf{t}^{(i)}$ | The ground-truth target for the $i$-th example |
| $\hat{\mathbf{y}}^{(i)}$ | Sampled target for the $i$-th example using model probabilities |
| $\xi$ | Source of training procedure randomness |
| $\xi_b$ | Randomness from batch ordering |
| $\Xi$ | A set containing various seeds $\xi$ |
| $\boldsymbol{\theta}$ | The neural network parameters |
| $\boldsymbol{\theta}^\star$ | Optimal parameters |
| $\boldsymbol{\theta}^\star(\mathcal{S})$ | Optimal parameters trained on data subset $\mathcal{S} \subseteq \mathcal{D}$ |
| $\boldsymbol{\theta}^s$ | Final model parameters (not necessarily at optimum) |
| $\boldsymbol{\theta}^s(\xi)$ | Final model parameters (not necessarily at optimum) which depends on randomness $\xi$ |
| $\boldsymbol{\theta}_k$ | The parameters of a network after $k$ iterations of an algorithm, which depends on context |
| $g(\boldsymbol{\theta}, \mathbf{x})$ | The output (logits) of a neural network with parameters $\boldsymbol{\theta}$ and input $\mathbf{x}$ |
| $\mathcal{L}(\mathbf{z}, \mathbf{t})$ | Loss function as a function of the neural network output and target (e.g., cross-entropy) |
| $\mathcal{L}(\boldsymbol{\theta}, z)$ | Loss function as a function of the parameters and training example |
| $\mathcal{J}(\boldsymbol{\theta}, \mathcal{D})$ | Cost function, $\mathcal{J}(\boldsymbol{\theta}, \mathcal{D}) = \frac{1}{N} \sum_{i=1}^N \mathcal{L}(\boldsymbol{\theta}, z_i)$ |
| $f_{z_q}(\boldsymbol{\theta})$ | Measurement function on query point $z_q$, typically correct-class margin or absolute error |
| $h_{f_{z_q}}(\boldsymbol{\theta})$ | The Neumann series iteration objective for the iHVP $\nabla_{\boldsymbol{\theta}} f_{z_q}(\boldsymbol{\theta}^s)^\top (\mathbf{G} + \lambda\mathbf{I})^{-1}$ |
| $h_{f_{z_q}}^{S_i}(\boldsymbol{\theta})$ | The Neumann series iteration objective projected onto the subspace corresponding to $S_i$ |
| $\mathbf{F}$ | The Fisher information matrix |
| $\mathbf{F}_{\text{EKFAC}}$ | The Fisher Information Matrix approximated with EKFAC |
| $\mathbf{G}$ | The Generalized Gauss-Newton Hessian (GGN) matrix |
| $\tilde{\mathbf{G}}_k$ | An unbiased sample of the GGN computed with data at iteration $k$ |
| $\mathbf{H}$ | Hessian matrix $\mathbf{H} := \nabla_{\boldsymbol{\theta}}^2 \mathcal{J}(\boldsymbol{\theta}^\star, \mathcal{D})$ |
| $\mathbf{P}$ | Preconditioning matrix used in ASTRA, chosen as the $\mathbf{G}_{\text{EKFAC}}$ |
| $\beta$ | Fraction of $\mathcal{D}$ used for subsampling when computing LDS |
| $\eta$ | Learning rate when training the neural network |
| $\alpha$ | Learning rate to find the iHVP with SNI or ASTRA |
| $\epsilon$ | Downweighting amount used in influence functions and SOURCE formulation |
| $\lambda$ | Damping parameter to compute iHVPs: a small positive scalar. |
| $\tilde{\lambda}$ | Damping parameter added to the preconditioner: a small positive scalar |
| $\tau$ | Training data attribution method |
| $\sigma$ | Eigenvalues in the decomposition of the curvature matrix. |
| $\Gamma$ | Group influence of a training data attribution method |
| $\rho$ | The Spearman correlation coefficient [37] |
| $\otimes$ | The Kronecker product |

## A.2 SOURCE specific notation

| Notation | Description |
|---|---|
| $\delta_{ki}$ | Indicator variable used in SOURCE formulation |
| $\ell$ | An index variable indicating the current segment in SOURCE |
| $K_\ell$ | The number of iterations within segment $\ell$ in SOURCE |
| $L$ | The number of segments in SOURCE |
| $T$ | The number of optimization steps for the underlying model |
| $\overline{\mathbf{g}}_\ell$ | The average gradient in segment $\ell$ |
| $\overline{\eta}_\ell$ | The average learning rate in segment $\ell$ |
| $\overline{\mathbf{r}}_\ell$ | Defined in Equation 20 |
| $\overline{\mathbf{S}}_\ell$ | Defined in Equation 20 |
| $\tilde{\mathbf{r}}_\ell$ | An approximation of $\overline{\mathbf{r}}_\ell$ introduced by Bae et al. [19] |

## A.3 Acronyms

| Acronym | Description |
|---|---|
| CG | Conjugate Gradient [51] |
| EKFAC | Eigenvalue-corrected Kronecker-factored Approximate Curvature [27] |
| FIM | Fisher Information Matrix |
| iHVP | Inverse Hessian-vector product, or inverse Gauss-Newton-Hessian-vector product |
| IF | Influence functions [17] |
| KFAC | Kronecker-factored Approximate Curvature [26] |
| LiSSA | Linear time Stochastic Second-Order Algorithm [23] |
| LOO | Leave-one-out |
| LDS | Linear Datamodeling Score [22] |
| MLP | Multi-layer perceptron |
| NI | Neumann series iterations |
| PBRF | Proximal-Bregman Response Function [36] |
| SGD | Stochastic gradient descent |
| SOURCE | Segmented statiOnary UnRolling for Counterfactual Estimation [19] |
| SNI | Stochastic Neumann series iterations |
| TDA | Training data attribution |

---

**Algorithm 1** iHVP approximation with LiSSA

---

**Require:** $\mathbf{v} \in \mathbb{R}^D$, $\alpha > 0$ (learning rate), $J > 0$ (number of iterations), $R > 0$ (repeat size), $\lambda > 0$ (damping term), $\mathcal{B} > 0$ (batch size), $\mathcal{D}$ (training dataset)

   $\mathbf{x} \leftarrow \mathbf{0}$            $\triangleright$ Initialize the accumulator for final estimation

   **for** $r = 1$ to $R$ **do**

      $\mathbf{v}_0 \leftarrow \mathbf{v}$          $\triangleright$ Initialize $\mathbf{v}_0$ as per the initial condition

      **for** $j = 0$ to $J - 1$ **do**

         $\mathcal{B} \leftarrow \text{SampleWithReplacement}(\mathcal{D}, \mathcal{B})$    $\triangleright$ Sample a mini-batch of size $\mathcal{B}$ from $\mathcal{D}$

         $\mathbf{p} \leftarrow \tilde{\mathbf{G}}_{\mathcal{B}} \mathbf{v}_j$         $\triangleright$ Compute HVP using mini-batch $\mathcal{B}$.

         $\mathbf{v}_{j+1} \leftarrow \mathbf{v}_j - \alpha(\mathbf{p} + \lambda \mathbf{v}_j) + \alpha \mathbf{v}$      $\triangleright$ SNI update rule

      **end for**

      $\mathbf{x} \leftarrow \mathbf{x} + \mathbf{v}_J$        $\triangleright$ Accumulate the result of this repetition

   **end for**

   $\mathbf{x} \leftarrow \mathbf{x}/R$         $\triangleright$ Average the accumulated results over $R$ repetitions

   **return** $\mathbf{x}$         $\triangleright$ Return final iHVP estimation $\mathbf{H}^{-1}\mathbf{v}$

---

## B  Extended Preliminaries

### B.1  LiSSA and SNI

LiSSA [23] is a second-order optimization algorithm which involves the computation of an iHVP. Koh and Liang [17] choose LiSSA as their iHVP solver, but LiSSA contains other components as well. In this paper, "LiSSA" refers to the iHVP component. Algorithm 1 shows that the primary difference between LiSSA and SNI is that the former repeats the SNI procedure multiple times to reduce variance (highlighted in red). We use $R = 1$ throughout this paper, so LiSSA's iHVP component is equivalent to SNI and thus we use the two terms interchangeably.

### B.2  Influence Functions

Previous papers have shown that influence function estimates are often fragile due to the strong assumptions in the influence function derivation [34–36, 42, 85, 86]. In Table 1, we outline the main assumptions.

Table 1: Summary of assumptions of Influence Functions vs. Unrolled Differentiation.

| Assumption | Influence Functions | Unrolled Differentiation |
|---|---|---|
| First order approximation | ✓ | ✓ |
| Objective convex with respect to parameters | ✓ | ✗ |
| Model trained to optimal parameters | ✓ | ✗ |

Despite the strong assumptions in the derivation of influence functions, a poor iHVP approximation can make influence functions estimates appear less reliable than they are. To appreciate these assumptions, we refer the reader to Appendix B.1 of [36] for a well-presented derivation of influence functions.

**Ensembling Influence Functions** TDA scores can be typically ensembled over multiple training trajectories [19, 22] to mitigate the problem of noise in the training procedure [42, 43]. This is typically done by training models with various seeds $\xi \in \Xi$, and approximating the *expected* first-order downweighting effect with the empirical average of attribution scores $\tau$:

$$\tau_{\text{IF-Ensemble}}(\boldsymbol{z}_m, \boldsymbol{z}_q, \mathcal{D}) := \frac{1}{|\Xi|} \sum_{\xi \in \Xi} \nabla_{\boldsymbol{\theta}} f_{\boldsymbol{z}_q}(\boldsymbol{\theta}^s(\xi))^{\top} (\mathbf{G}_{\xi} + \lambda \mathbf{I})^{-1} \nabla_{\boldsymbol{\theta}} \mathcal{L}(\boldsymbol{\theta}^s(\xi), \boldsymbol{z}_m), \quad (15)$$

where $\mathbf{G}_{\xi}$ is the GGN computed at $\boldsymbol{\theta}^s(\xi)$. We perform ensembling in this paper using the procedure in Equation 15, and apply ensembling for SOURCE analogously. Note that Equation 15 ensembles the attribution scores [19, 22], while some other works ensemble the weights [19, 87].

## B.3 Curvature Matrices

We explain the relationships between curvature matrices below for completeness, which is heavily based on Grosse [49] and Martens [55].

**Approximating H with G** Throughout this paper, we use the approximation $\mathbf{G} \approx \mathbf{H}$. Letting $\mathbf{z} = g(\boldsymbol{\theta}, \mathbf{x})$ denote the neural network output[17], the GGN is equal to the Hessian if we drop the second term from the following decomposition of $\mathbf{H}$ [49]:

$$\mathbf{H} = \frac{1}{N} \sum_{(\mathbf{x}^{(i)}, \mathbf{t}^{(i)}) \in \mathcal{D}} [\mathbf{J}_{\mathbf{z}^{(i)}\boldsymbol{\theta}}^{\top} \mathbf{H}_{\mathbf{z}^{(i)}} \mathbf{J}_{\mathbf{z}^{(i)}\boldsymbol{\theta}} + \sum_{j} \frac{\partial \mathcal{L}}{\partial \mathbf{z}_{j}^{(i)}} \nabla_{\boldsymbol{\theta}}^{2} g(\boldsymbol{\theta}, \mathbf{x}^{(i)})_{j}], \tag{16}$$

where $\mathbf{J}_{\mathbf{z}^{(i)}\boldsymbol{\theta}}$ is the Jacobian matrix of the neural network's outputs with respect to the parameters for the $i$th training example, $\mathbf{H}_{\mathbf{z}^{(i)}} := \nabla_{\mathbf{z}}^{2}\mathcal{L}(\mathbf{z}^{(i)}, \mathbf{t}^{(i)})$ refers to the Hessian of the loss function with respect to the neural network outputs for the $i$th training example, $\frac{\partial \mathcal{L}}{\partial \mathbf{z}_{j}^{(i)}}$ refers to the derivative of the loss function with respect to the $j$th neural network output for the $i$th training example and $\nabla_{\boldsymbol{\theta}}^{2} g(\boldsymbol{\theta}, \mathbf{x}^{(i)})_{j}$ refers to the Hessian of the $j$th neural network output for the $i$th training example with respect to the parameters. For a linear neural network, $\nabla_{\boldsymbol{\theta}}^{2} g(\boldsymbol{\theta}, \mathbf{x}^{(i)})_{j} = 0$, so the GGN is equal to the Hessian if we linearize the neural networks with respect to the parameters and only capture the curvature in the loss function. Linearization of the neural network is an approximation documented in previous works [55, 88–90] and used frequently for influence functions if the damped GGN $\mathbf{G} + \lambda \mathbf{I}$ is used [7, 11, 22, 36].

**Equivalence of F and G** For the machine learning tasks that we consider, such as regression and classification tasks, the outputs of the neural network can be seen as the natural parameters of an exponential family. For these cases, the Fisher Information Matrix and the GGN coincide (i.e. $\mathbf{F} = \mathbf{G}$). We will illustrate this for softmax classification, but the case for regression can be derived similarly. The cross-entropy loss for a training example $z = (\mathbf{x}, \mathbf{t})$ is $\mathcal{L}(\boldsymbol{\theta}, z) = -\mathbf{t}^{\top} \log p_{\boldsymbol{\theta}}(\mathbf{y}|\mathbf{x})$ whose gradient is: $\nabla_{\boldsymbol{\theta}} \mathcal{L}(\boldsymbol{\theta}, z) = \mathbf{J}_{\mathbf{z}\boldsymbol{\theta}}^{\top} (p_{\boldsymbol{\theta}}(\mathbf{y}|\mathbf{x}) - \mathbf{t})$. Then the following equalities hold:

$$\mathbf{F} = \frac{1}{N} \sum_{(\mathbf{x}^{(i)}, \mathbf{t}^{(i)}) \in \mathcal{D}} \mathbb{E}_{\hat{\mathbf{y}} \sim p_{\boldsymbol{\theta}}(\mathbf{y}|\mathbf{x}^{(i)})} \left[ \nabla_{\boldsymbol{\theta}} \log p_{\boldsymbol{\theta}}(\hat{\mathbf{y}}|\mathbf{x}^{(i)}) \nabla_{\boldsymbol{\theta}} \log p_{\boldsymbol{\theta}}(\hat{\mathbf{y}}|\mathbf{x}^{(i)})^{\top} \right]$$

$$= \frac{1}{N} \sum_{(\mathbf{x}^{(i)}, \mathbf{t}^{(i)}) \in \mathcal{D}} \mathbb{E}_{\hat{\mathbf{y}} \sim p_{\boldsymbol{\theta}}(\mathbf{y}|\mathbf{x}^{(i)})} \left[ \mathbf{J}_{\mathbf{z}^{(i)}\boldsymbol{\theta}}^{\top} (p_{\boldsymbol{\theta}}(\hat{\mathbf{y}} \mid \mathbf{x}^{(i)}) - \hat{\mathbf{y}})(p_{\boldsymbol{\theta}}(\hat{\mathbf{y}} \mid \mathbf{x}^{(i)}) - \hat{\mathbf{y}})^{\top} \mathbf{J}_{\mathbf{z}^{(i)}\boldsymbol{\theta}} \right]$$

$$= \frac{1}{N} \sum_{(\mathbf{x}^{(i)}, \mathbf{t}^{(i)}) \in \mathcal{D}} \mathbf{J}_{\mathbf{z}^{(i)}\boldsymbol{\theta}}^{\top} \left( \mathrm{diag}(p_{\boldsymbol{\theta}}(\hat{\mathbf{y}} \mid \mathbf{x}^{(i)})) - p_{\boldsymbol{\theta}}(\hat{\mathbf{y}} \mid \mathbf{x}^{(i)}) p_{\boldsymbol{\theta}}(\hat{\mathbf{y}} \mid \mathbf{x}^{(i)})^{\top} \right) \mathbf{J}_{\mathbf{z}^{(i)}\boldsymbol{\theta}}$$

$$= \frac{1}{N} \sum_{(\mathbf{x}^{(i)}, \mathbf{t}^{(i)}) \in \mathcal{D}} \mathbf{J}_{\mathbf{z}^{(i)}\boldsymbol{\theta}}^{\top} \nabla_{\mathbf{z}}^{2} \mathcal{L}(\mathbf{z}^{(i)}, \mathbf{t}^{(i)}) \mathbf{J}_{\mathbf{z}^{(i)}\boldsymbol{\theta}} = \mathbf{G}$$

## B.4 Training Data Attribution with Unrolled Differentiation

Influence functions may struggle with models that have not converged [19, 35, 36, 83] (Table 1). Fortunately, unrolled differentiation methods [4, 18–21] do not rely on model convergence. Instead, they capture the effect of downweighting a training example by differentiating through the entire training trajectory. They can also capture the effect of other sources of randomness, such as batch ordering [41]. Assume our optimization algorithm is mini-batch gradient descent, which uses a learning rate $\eta_k$ and batch size $B$, and let $\delta_{ki}$ be an indicator variable that equals 1 if and only if $z_m = z_{ki}$, where $z_{ki}$ is the $i$-th training example in batch $k$.[18] Then the mini-batch gradient descent

---

[17]Note the difference between the bold font $\mathbf{z}$, which refers to neural network outputs, with italicized font $z$, which refers to a data point $z = (\mathbf{x}, \mathbf{t})$

[18]We note that $\boldsymbol{\theta}_k$ in this setting refers to the parameters at time step $k$ when training the network and distinguish it from SNI or ASTRA iterations presented in Section 2 and Section 3.

update rule is:

$$\boldsymbol{\theta}_{k+1}(\epsilon) \leftarrow \boldsymbol{\theta}_k(\epsilon) - \frac{\eta_k}{B}\sum_{i=1}^{B}(1-\delta_{ki}\epsilon)\nabla_{\boldsymbol{\theta}}\mathcal{L}(\boldsymbol{\theta}_k(\epsilon), \boldsymbol{z}_{ki}), \tag{17}$$

where $\delta_k := \sum_{i=1}^{B}\delta_{ki}$. Let $\xi_b$ denote the randomness from batch ordering. Then the expected first-order effect on the parameters $\boldsymbol{\theta}_T$ after $T$ steps of training with $\boldsymbol{z}_m$ downweighted by $\epsilon$ is:[19]

$$\mathbb{E}_{\xi_b}\left[\frac{\mathrm{d}\boldsymbol{\theta}_T}{\mathrm{d}\epsilon}\right] = -\mathbb{E}_{\xi_b}\left[\sum_{k=0}^{T-1}\frac{\eta_k}{B}\delta_k\left(\prod_{i=T-1}^{k+1}(\mathbf{I}-\eta_i\mathbf{G}_i)\right)\nabla_{\boldsymbol{\theta}}\mathcal{L}(\boldsymbol{\theta}_k, \boldsymbol{z}_m)\right], \tag{18}$$

Bae et al. [19] introduce an algorithm called SOURCE, which approximates Equation 18 much more cheaply by segmenting the trajectory into $L$ segments, assuming stationary and independent GGNs and gradients within each segment. For the $\ell$th segment which starts at iteration $T_{\ell-1}$ and ends at iteration $T_\ell$, and $k$ satisfying $T_{\ell-1} \leq k < T_\ell$, let $\overline{\mathbf{G}}_\ell := \mathbb{E}_{\xi_b}[\mathbf{G}_k]$, $\overline{\mathbf{g}}_\ell := \mathbb{E}_{\xi_b}[\nabla_{\boldsymbol{\theta}}\mathcal{L}(\boldsymbol{\theta}_k, \boldsymbol{z}_m)]$, and $\overline{\eta}_\ell$ refer to the average GGN, average gradient, and average learning rate in segment $\ell$ respectively and let $K_\ell := T_\ell - T_{\ell-1}$ refer to the total number of iterations in segment $\ell$. Then SOURCE approximates the first-order effect of downweighting parameters as:

$$\mathbb{E}_{\xi_b}\left[\frac{\mathrm{d}\boldsymbol{\theta}_T}{\mathrm{d}\epsilon}\right] \approx -\frac{1}{N}\sum_{\ell=1}^{L}\left(\prod_{\ell'=L}^{\ell+1}(\mathbf{I}-\overline{\eta}_{\ell'}\overline{\mathbf{G}}_{\ell'})^{K_{\ell'}}\right)\left(\sum_{k=T_{\ell-1}}^{T_\ell-1}\overline{\eta}_\ell(\mathbf{I}-\overline{\eta}_\ell\overline{\mathbf{G}}_\ell)^{T_\ell-1-k}\overline{\mathbf{g}}_\ell\right) \tag{19}$$

$$\approx -\frac{1}{N}\sum_{\ell=1}^{L}\left(\prod_{\ell'=L}^{\ell+1}\underbrace{\exp(-\overline{\eta}_{\ell'}K_{\ell'}\overline{\mathbf{G}}_{\ell'})}_{\overline{\mathbf{S}}_{\ell'}}\right)\underbrace{\left(\mathbf{I}-\exp(-\overline{\eta}_\ell K_\ell\overline{\mathbf{G}}_\ell)\right)\overline{\mathbf{G}}_\ell^{-1}\overline{\mathbf{g}}_\ell}_{\overline{\mathbf{r}}_\ell}. \tag{20}$$

The stationary and independent assumptions allow us to factor shared products resulting in the approximation in Equation 19, which can be then approximated with matrix exponentials in Equation 20 and computed with the EKFAC eigendecomposition. Bae et al. [19] provide an interpretation of the $\overline{\mathbf{r}}_\ell$ term, noticing that $\overline{\mathbf{r}}_\ell \approx \left(\overline{\mathbf{G}}_\ell + \overline{\eta}_\ell^{-1}K_\ell^{-1}\mathbf{I}\right)^{-1}\overline{\mathbf{g}}_\ell := \tilde{\mathbf{r}}_\ell$, which is an iHVP that we can use to apply ASTRA. We discuss this term in more detail in Appendix C. We refer the reader to Bae et al. [19] for a full explanation of SOURCE.

## B.5 Kronecker-Factored Approximate Curvature

The KFAC approximation was introduced in [26] in the context of second-order optimization and explained in [7] in the context of influence functions. To understand the cost of EKFAC and EKFAC-IF compared to ASTRA-IF, as well as the assumptions EKFAC make which results in a biased iHVP approximation, we present the derivation below which is heavily based on Grosse [49] and Grosse et al. [7] and refer readers to Martens and Grosse [26] and George et al. [27] for further reading.

Our goal is to compute the iHVP with the Fisher Information Matrix (FIM) $\mathbf{F}$ as an approximation for the Hessian $\mathbf{H}$. The FIM is defined as:

$$\mathbf{F} := \frac{1}{N}\sum_{\mathbf{x}^{(i)}\in\mathcal{D}}\mathbb{E}_{\hat{\mathbf{y}}\sim p_{\boldsymbol{\theta}}(\mathbf{y}|\mathbf{x}^{(i)})}\left[\nabla_{\boldsymbol{\theta}}\log p_{\boldsymbol{\theta}}(\hat{\mathbf{y}}|\mathbf{x}^{(i)})\nabla_{\boldsymbol{\theta}}\log p_{\boldsymbol{\theta}}(\hat{\mathbf{y}}|\mathbf{x}^{(i)})^\top\right] \tag{21}$$

where $p_{\boldsymbol{\theta}}(\mathbf{y}|\mathbf{x})$ is the model's own distribution over targets. We omit the random variables in the expectation's subscripts going forward to reduce clutter. Using the model's own distribution over targets (as opposed to actual targets) is rather important since using the actual targets rather than the model's distribution results in a matrix called the Empirical Fisher, which does not have the same properties as the FIM [91].

We now describe KFAC for multilayer perceptrons. We refer readers to Grosse and Martens [84] for the derivation for convolution layers. Consider the $l$-th layer of a neural network,[20] which has input

---

[19]Unless otherwise stated, derivatives taken with respect to $\epsilon$ are evaluated at $\epsilon = 0$. The notation $\prod_{i=T-1}^{k+1}$ means taking products in decreasing order from $T-1$ to $k+1$.

[20]Note the difference between $\ell$, which refers to a segment in SOURCE, and $l$, which is an index denoting a layer of a neural network.

activations $\mathbf{a}_{l-1} \in \mathbb{R}^I$, weights $\mathbf{W}_l \in \mathbb{R}^{O \times I}$, bias $\mathbf{b}_l \in \mathbb{R}^O$, and outputs $\mathbf{s}_l \in \mathbb{R}^O$. For convenience, we use the notation $\bar{\mathbf{a}}_{l-1} = \begin{bmatrix} \mathbf{a}_{l-1}^\top & 1 \end{bmatrix}^\top$ and $\overline{\mathbf{W}}_l = [\mathbf{W}_l \; \mathbf{b}_l]$ to handle weights and biases together, and we write $\boldsymbol{\theta}_l = \text{vec}(\overline{\mathbf{W}}_l)$ to denote the reshaped vector of layer $l$ parameters. Then each layer computes:

$$\mathbf{s}_l = \overline{\mathbf{W}}_l \, \bar{\mathbf{a}}_{l-1}, \quad \mathbf{a}_l = \phi_l(\mathbf{s}_l), \tag{22}$$

where $\phi_l$ is the activation function. We will define the *pseudo-gradient* operator as:

$$\mathcal{D}v := \nabla_v \log p_{\boldsymbol{\theta}}(\hat{\mathbf{y}} \mid \mathbf{x}) \tag{23}$$

for notational convenience. Notice that $\mathcal{D}v$ is a random variable whose randomness arises from sampling $\hat{\mathbf{y}}$. Using the properties of the Kronecker product, we can write the pseudo-gradient of $\boldsymbol{\theta}_l$ as:

$$\mathcal{D}\boldsymbol{\theta}_l = \text{vec}(\mathcal{D}\overline{\mathbf{W}}_l) = \text{vec}(\mathcal{D}\mathbf{s}_l \bar{\mathbf{a}}_{l-1}^\top) = \bar{\mathbf{a}}_{l-1} \otimes \mathcal{D}\mathbf{s}_l, \tag{24}$$

where $\otimes$ is the Kronecker product. Then the $l$th block of the FIM can be approximated as:

$$\mathbf{F}_l = \mathbb{E}\left[\mathcal{D}\boldsymbol{\theta}_l \mathcal{D}\boldsymbol{\theta}_l^\top\right] \tag{25}$$

$$= \mathbb{E}\left[(\bar{\mathbf{a}}_{l-1} \otimes \mathcal{D}\mathbf{s}_l)(\bar{\mathbf{a}}_{l-1} \otimes \mathcal{D}\mathbf{s}_l)^\top\right] \tag{26}$$

$$= \mathbb{E}\left[\bar{\mathbf{a}}_{l-1} \bar{\mathbf{a}}_{l-1}^\top \otimes \mathcal{D}\mathbf{s}_l \mathcal{D}\mathbf{s}_l^\top\right] \tag{27}$$

$$\approx \mathbb{E}\left[\bar{\mathbf{a}}_{l-1} \bar{\mathbf{a}}_{l-1}^\top\right] \otimes \mathbb{E}\left[\mathcal{D}\mathbf{s}_l \mathcal{D}\mathbf{s}_l^\top\right] \tag{28}$$

$$= \mathbf{A}_{l-1} \otimes \mathbf{S}_l, \tag{29}$$

where have applied Kronecker product identities on the third equality. Our final approximation $\widehat{\mathbf{F}}$ to the FIM is the block-diagonal matrix in which each block is $\mathbf{F}_l$. Here, $\mathbf{A}_{l-1} = \mathbb{E}[\bar{\mathbf{a}}_{l-1} \bar{\mathbf{a}}_{l-1}^\top]$ and $\mathbf{S}_l = \mathbb{E}[\mathcal{D}\mathbf{s}_l \mathcal{D}\mathbf{s}_l^\top]$ are the uncentered covariance matrices of the activations and the pre-activation pseudo-gradients with dimensions $(I+1) \times (I+1)$ and $O \times O$, respectively. Practically, we can estimate the expectations via an empirical estimate and store the resulting statistics $\widehat{\mathbf{A}}_{l-1} = \frac{1}{N} \sum_{\mathcal{D}} \bar{\mathbf{a}}_{l-1} \bar{\mathbf{a}}_{l-1}^\top$ and $\widehat{\mathbf{S}}_l = \frac{1}{N} \sum_{\mathcal{D}} \mathcal{D}\mathbf{s}_l \mathcal{D}\mathbf{s}_l^\top$, and we define $\widehat{\mathbf{F}}_l := \widehat{\mathbf{A}}_{l-1} \otimes \widehat{\mathbf{S}}_l$.

To approximate $(\mathbf{F} + \lambda\mathbf{I})^{-1}\mathbf{v}$ for a vector $\mathbf{v}$ as needed to compute influence functions, we can compute $(\mathbf{F}_l + \lambda\mathbf{I})^{-1}\mathbf{v}_l$ separately for each layer $l$. Let $\overline{\mathbf{V}}_l$ be the slice of $\mathbf{v}$ reshaped to match $\overline{\mathbf{W}}_l$, and define $\mathbf{v}_l = \text{vec}(\overline{\mathbf{V}}_l)$. Applying the eigendecompositions $\mathbf{A}_{l-1} = \mathbf{Q}_{\mathbf{A}_{l-1}} \mathbf{D}_{\mathbf{A}_{l-1}} \mathbf{Q}_{\mathbf{A}_{l-1}}^\top$ and $\mathbf{S}_l = \mathbf{Q}_{\mathbf{S}_l} \mathbf{D}_{\mathbf{S}_l} \mathbf{Q}_{\mathbf{S}_l}^\top$, and using the Kronecker identity $\mathbf{U} \otimes \mathbf{V} = (\mathbf{Q}_\mathbf{U} \otimes \mathbf{Q}_\mathbf{V})(\mathbf{D}_\mathbf{U} \otimes \mathbf{D}_\mathbf{V})(\mathbf{Q}_\mathbf{U}^\top \otimes \mathbf{Q}_\mathbf{V}^\top)$ for two symmetric matrices $\mathbf{U} \otimes \mathbf{V}$, we can write:

$$(\mathbf{F}_l + \lambda\mathbf{I})^{-1}\mathbf{v}_l = (\mathbf{A}_{l-1} \otimes \mathbf{S}_l + \lambda\mathbf{I})^{-1}\mathbf{v}_l \tag{30}$$

$$= (\mathbf{Q}_{\mathbf{A}_{l-1}} \otimes \mathbf{Q}_{\mathbf{S}_l}) \big( \underbrace{\mathbf{D}_{\mathbf{A}_{l-1}} \otimes \mathbf{D}_{\mathbf{S}_l}}_{\text{Scaling matrix } \Lambda_{\text{KFAC}}} + \lambda\mathbf{I}_{\mathbf{A}_{l-1}} \otimes \mathbf{I}_{\mathbf{S}_l} \big)^{-1} \underbrace{(\mathbf{Q}_{\mathbf{A}_{l-1}} \otimes \mathbf{Q}_{\mathbf{S}_l})^\top}_{\text{Orthonormal eigenbasis}} \mathbf{v}_l \tag{31}$$

where $\mathbf{I}_{\mathbf{A}_{l-1}}$ and $\mathbf{I}_{\mathbf{S}_l}$ represent identity matrices of the same shape as $\mathbf{A}_{l-1}$ and $\mathbf{S}_l$ respectively.

**Eigenvalue Corrected Kronecker-Factored Approximate Curvature**   One simple adjustment to the KFAC approximation can yield material improvements to the curvature approximation as well as the influence approximation. The KFAC formulation in Equation 31 suggests that after expressing $\mathbf{v}_l$ in the eigenbasis $(\mathbf{Q}_{\mathbf{A}_{l-1}} \otimes \mathbf{Q}_{\mathbf{S}_l})^\top$ we scale each element with $(\mathbf{D}_{\mathbf{A}_{l-1}} \otimes \mathbf{D}_{\mathbf{S}_l} + \lambda\mathbf{I}_{\mathbf{A}_{l-1}} \otimes \mathbf{I}_{\mathbf{S}_l})^{-1}$. Observing that for any matrix $\mathbf{W} = \mathbb{E}[\mathbf{u}\mathbf{u}^\top] = \mathbf{U}\mathbf{S}\mathbf{U}^\top$, it is true that $\mathbf{S}_{ii} = \mathbb{E}[(\mathbf{U}^\top\mathbf{v})_i^2]$, George et al. [27] propose that a better scaling matrix is the diagonal matrix $\Lambda_{\text{EKFAC}}$ with entries:

$$\Lambda_{ii} = \mathbb{E}\big[\big((\mathbf{Q}_{\mathbf{A}_{l-1}} \otimes \mathbf{Q}_{\mathbf{S}_l})^\top \mathcal{D}\boldsymbol{\theta}_l\big)_i^2\big]. \tag{32}$$

In practice, this eigenvalue correction results in better influence estimates compared with KFAC.

**Assumptions in EKFAC**   From the equations above, we can see that KFAC makes two critical approximations which EKFAC inherits: First, it assumes that correlations between $\mathcal{D}\boldsymbol{\theta}_i$ and $\mathcal{D}\boldsymbol{\theta}_j$ are zero if they belong to different layers, yielding a block-diagonal approximation of $\mathbf{F}$. Second, it treats the activations as independent from the pre-activation pseudo-gradients, the basis of Equation 28.

Furthermore, the matrix $\mathbf{S}_l$ depends on the *sampled* labels $\widehat{\mathbf{y}}$, which for efficiency reasons is usually only sampled once per input $\mathbf{x}$ which also may introduce some approximation error. In total, these assumptions cause $\mathbf{F}_{\text{EKFAC}}$ to differ from the true FIM $\mathbf{F}$, which is an error that ASTRA corrects. Grosse and Martens [84] introduce the KFAC approximation for convolution layers which introduces two further assumptions – spatially uncorrelated derivatives and spatial homogeneity. Consistent with past works [1, 19], we find that EKFAC-IF struggles for convolution architectures in comparison with MLPs. ASTRA-IF's TDA performance on ResNet-9 achieves a particularly large improvement compared to EKFAC-IF.

**Cost of Computing EKFAC-IF**  The three main components of computing EKFAC-IF are: 1) collecting the statistics $\widehat{\mathbf{A}}_{l-1}$ and $\widehat{\mathbf{S}}_l$, which requires a backward pass over the whole dataset. 2) computing the eigendecompositions $\mathbf{A}_{l-1} = \mathbf{Q}_{\mathbf{A}_{l-1}} \mathbf{D}_{\mathbf{A}_{l-1}} \mathbf{Q}_{\mathbf{A}_{l-1}}^\top$ and $\mathbf{S}_l = \mathbf{Q}_{\mathbf{S}_l} \mathbf{D}_{\mathbf{S}_l} \mathbf{Q}_{\mathbf{S}_l}^\top$, whose total precise cost depends on the architecture [7]. 3) if we want to search the whole dataset $\mathcal{D}$ for influential examples, once we approximate $(\mathbf{F} + \lambda \mathbf{I})^{-1} \nabla_{\boldsymbol{\theta}} f_{z_q}(\boldsymbol{\theta}^s)$ via the procedure above, we need to take a dot product with every training example gradient under consideration in $\mathcal{D}$. ASTRA adds an incremental iterative procedure to the cost of EKFAC-IF which results in stronger TDA performance.

# C  Introducing ASTRA-SOURCE

**Understanding the iHVP**  We have discussed how to compute iHVPs and therefore influence functions with ASTRA. We can also apply ASTRA to SOURCE by making the substitution $\bar{\mathbf{r}}_\ell \approx \left( \overline{\mathbf{G}}_\ell + \bar{\eta}_\ell^{-1} K_\ell^{-1} \mathbf{I} \right)^{-1} \overline{\mathbf{g}}_\ell := \tilde{\mathbf{r}}_\ell$ described in Appendix B.4, which replaces the matrix exponential in $\bar{\mathbf{r}}_\ell$. To understand the approximation, observe that the following term (with some rearranging) found in Equation 19 $\bar{\eta}_\ell \sum_{k=T_{\ell-1}}^{T_\ell - 1} (\mathbf{I} - \bar{\eta}_\ell \overline{\mathbf{G}}_\ell)^{T_\ell - 1 - k} \overline{\mathbf{g}}_\ell$ resembles applying the Neumann series iterations on the vector $\mathbf{v} = \overline{\mathbf{g}}_\ell$ and the matrix $\mathbf{A} = \overline{\mathbf{G}}_\ell$ with a learning rate of $\bar{\eta}_\ell$ for a total of $K_\ell - 1$ iterations. For large enough $K_\ell$, the truncation can be seen as approximately running Neumann series iterations until convergence, which results in the iHVP $\overline{\mathbf{G}}_\ell^{-1} \overline{\mathbf{g}}_\ell$. However, each segment actually only involves $K_\ell$ iterations at an average learning rate of $\bar{\eta}_\ell$, and thus is more closely related to *truncated* Neumann series iterations (Appendix G), in which the parameters make less progress in the low eigenvalue directions. Bae et al. [19] provide an interpretation that this can be approximated with damping as follows: $\bar{\eta}_\ell \sum_{k=T_{\ell-1}}^{T_\ell - 1} (\mathbf{I} - \bar{\eta}_\ell \overline{\mathbf{G}}_\ell)^{T_\ell - 1 - k} \overline{\mathbf{g}}_\ell \approx (\overline{\mathbf{G}} + \bar{\eta}_\ell^{-1} K_\ell^{-1} \mathbf{I})^{-1} \overline{\mathbf{g}}_\ell$, and that over a wide range of eigenvalues, the qualitative behavior matches well. The transformation of this term into an iHVP allows us to apply ASTRA to SOURCE.

**Practical Implementation**  Recall that the final goal in SOURCE is to approximate $\nabla_{\boldsymbol{\theta}} f_{z_q}(\boldsymbol{\theta}^s)^\top \mathbb{E}_{\xi_b} \left[ \frac{d\boldsymbol{\theta}_T}{d\epsilon} \right]$. To do this, we follow Equation 20 from left to right: for all $\ell = 1, \dots, L$ segments, we first compute $-\frac{1}{N} \nabla_{\boldsymbol{\theta}} f_{z_q}(\boldsymbol{\theta}^s)^\top \prod_{\ell'=L}^{\ell+1} \overline{\mathbf{S}}_{\ell'}$ in the same manner as SOURCE. This will be the vector in our iHVP. ASTRA differs from SOURCE in that instead of multiplying this vector by another matrix exponential, we multiply it by the inverse damped GGN $\left( \overline{\mathbf{G}}_\ell + \bar{\eta}_\ell^{-1} K_\ell^{-1} \mathbf{I} \right)^{-1}$, which we can do using ASTRA in the same manner described previously. Finally, we multiply by the average gradient, $\overline{\mathbf{g}}_\ell$ and accumulate the result over $L$ segments, which is done in the same manner as SOURCE. Compared to ASTRA-IF, there is an additional detail introduced: how to obtain the average GGN $\overline{\mathbf{G}}_\ell$. There are a number of options available, but we find that using the average weights in the segment works well, which we discuss below. The full procedure of applying ASTRA to SOURCE requires $L$ iHVPs per query. In many cases, the number of segments $L$ is likely to be small.[21] Since the preconditioners used by ASTRA would need to be computed if we were to run EKFAC-SOURCE anyways, the incremental cost of ASTRA-SOURCE compared to EKFAC-SOURCE is no more than $L$ times the number of iterations for each iHVP, which is hundreds of iterations in our experiments.

**Approximation of Other Terms**  We have discussed how to improve the approximation of $\bar{\mathbf{r}}_\ell$ with ASTRA, but have not discussed $\overline{\mathbf{S}}_{\ell'}$. We found evidence that improving the quality of the term $\bar{\mathbf{r}}_\ell$ improved TDA performance, but could not find the same evidence for $\overline{\mathbf{S}}_{\ell'}$. Therefore we spent

---

[21]Part of the motivation for SOURCE is to devise a scalable TDA algorithm for multi-stage training procedures, in which the number of segments is likely to be modest. Bae et al. [19] use $L = 3$.

additional compute on improving the iHVP approximation. As a result, we will leave $\overline{\mathbf{S}}_{\ell'}$ as the approximation involving EKFAC.

**Computing the Average Gauss-Newton Hessian**    There are a number of options in computing the average GGN $\overline{\mathbf{G}}_\ell$. **Option 1**): since $\overline{\mathbf{G}}_\ell := \mathbb{E}_{\xi_b}[\mathbf{G}_k]$ for $T_{\ell-1} \leq k < T_\ell$, one can compute the matrix-vector product involving $\overline{\mathbf{G}}_\ell$ and $\mathbf{v} \in \mathbb{R}^D$ simply by taking an empirical average over samples within the segment: $\overline{\mathbf{G}}_\ell \mathbf{v} \approx \frac{1}{K_\ell} \sum_{T_{\ell-1} \leq k < T_\ell} \mathbb{E}_{\xi_b}[\mathbf{G}_k]\mathbf{v}$. This might be costly since one would have to load multiple checkpoints into memory just to compute a forward pass. **Option 2**): Instead of sampling *every* checkpoint in the segment, we could reduce the number of samples and take the empirical average of that instead. This is consistent with SOURCE as it uses only a subset of checkpoints in each segment anyways. If the segments chosen in SOURCE are indeed stationary as the derivation approximates, then in the extreme, we could take one checkpoint in each segment as the representative. **Option 3**): Bae et al. [19] provides an alternative averaging scheme called FAST-SOURCE, in which one averages the parameters rather than the gradients, and shows that it works approximately as well as SOURCE. We tested Option 2 and Option 3 for a few settings and could not find meaningful differences, so opted to present the results for Option 3, using the average weights.

# D    Extended Related Works

**Understanding Influence Functions**    A number of previous works [34–36, 42, 85, 86, 92] have studied influence functions accuracy in various modern neural network settings. [34] show that influence functions do not approximate LOO retraining well. [36] discover that the derivation of influence functions actually approximate the *Proximal Bregman Response Function* (PBRF) rather than LOO retraining, which can be seen as an objective which tries to *maximize* loss on the removed training example, subject to constraints in function space and weight space measured from the final parameters. Two large contributions to influence functions error are the *warm-start retraining* assumption, which assumes that the counterfactual model is initialized at the final parameters, and the *non-convergence gap*, which relates to the fact that the derivation assumes the model has converged to an optimal solution. Ensembling can help address the warm-start retraining assumption, while the non-convergence gap is addressed by [19]. Others have focused on whether influence functions can accurately approximate group influence [5, 10, 92, 93] as it makes a linearity assumption due to the fact that it is a first-order approximation. While LOO influence is very noisy [19, 43], Ilyas et al. [10] discover that model predictions are approximately linear with respect to training example inclusion, which provides the justification for LDS [22]. Overall, these works typically aim to address the fundamental assumptions surrounding influence functions (Table 1). In contrast, our work shows that a poorly approximated iHVP can cause substantial performance degradation.

**Ensembling in Influence Functions**    Ensembling combines multiple models for improved generalization, uncertainty estimation, and calibration. It is a common approach to estimate the model posterior $p(\boldsymbol{\theta}|\mathcal{D})$ in Bayesian deep learning. Different strategies for sampling models as members of an ensemble exist: For example, deep ensembles sample models from varying random initializations [94] to represent variations stemming from possible training trajectories, while stochastic weight averaging (SWA) approaches sample model parameters from the final iterations of model training [95, 96], which has the advantage of reduced training cost. Other methods may combine these ideas [97], or use different approaches (e.g., Dropout [98]). Taking the average across several runs with varying sources of training process randomness is a common approach to account for the variability of model training in TDA estimation [10, 19, 22]. This can be seen as sampling from the distribution of true TDA to estimate the average treatment effect of excluding a training subset [43] and has been effective in stabilizing estimations as well as evaluations (e.g. the LDS [22]). The size of the ensemble is generally connected to improved estimation quality of TDA scores, as shown in [22]. Our results show that ASTRA enjoys a larger performance boost from ensembling.

# E    Evaluating TDA

In this paper, we evaluate the performance of TDA methods with the LDS [22], a widely used metric which we shortly described in Section 5. Besides LDS, other evaluations for measuring TDA performance also exist. In this section, we present alternate methods of TDA evaluation.

**Expected leave-one-out retraining.**    TDA methods usually define the influence of a training sample $z_m$ on the model as the change in the model's predictions if $z_m$ were not part of the training set. Hence, a straightforward way to compute the ground-truth to compare against is leave-one-out (LOO) retraining, as done in [6, 17, 99]. However, since the stochasticity inherent to model training makes LOO a noisy measure [42, 43], the LOO score should be considered in expectation over the training process stochasticity $\xi$. We can then define the expected leave-one-out (ELOO) score as:

$$\mathrm{ELOO}(z_m, z_q, \mathcal{D}) := \mathbb{E}_\xi[f_{z_q}(\boldsymbol{\theta}^s(\mathcal{D} \setminus \{z_m\}; \xi))] - \mathbb{E}_\xi[f_{z_q}(\boldsymbol{\theta}^s(\mathcal{D}; \xi))]. \tag{33}$$

This score can be viewed as the ground-truth average treatment effect (ATE) [100] of the removal of $z_m$ from $\mathcal{D}$. While principled, in practice, the effect of removing a single point is highly noisy [19, 43] so that a stable estimate of ELOO may only be achieved with an extremely large number of samples to compute the empirical expectation. In contrast, LDS considers the ATE of excluding a *group* of training samples from training, which has been shown to be more stable in expectation [19, 22].

**Top-k removal and retraining.**    The sign of the TDA scores indicates whether the excluded training samples are positively or negatively influential [19], also referred to as proponents and opponents [83], helpful and harmful samples [17], or excitatory and inhibitory [6], respectively [19]. The idea behind this evaluation is that the removal of positively influential samples $\{z_m\}$ removes support for the query sample $z_q$ and consequently should lead to a change in prediction confidence on $z_q$ [101]. [101] conduct this evaluation by removing the top and bottom 10% of training samples ranked by influence functions and compare against the removal of the least influential (i.e., smallest influence scores by magnitude) and random samples to see if the resulting models change their predictions in the expected way. Similar to top-k removal and retraining, previous work has tested how many highly influential samples need to be removed to *flip* a prediction [19, 102], called subset removal counterfactual evaluation. Similar to LDS, removing top-k samples is based on counterfactual retraining with excluded groups. However, one core difference is that since LDS involves a sum over all the attribution scores for every $z_m$ in each subset (Equation 10), poorly *calibrated* attribution scores resulting in one outlier score $\tau(z_m, z_q, \mathcal{D})$ may result in poor LDS, demanding that the TDA method assigns calibrated scores across all points $z_m$ in consideration.

# F    Experiment Details

## F.1    Choice of Measurement Function

While in principle the derivation of our influence scores does not restrict what measurement function $f_{z_q}$ is used, in practice some choices of measurement functions work better than others. In our experiments, for regression problems, we use the measurement function:

$$f_{z_q}(\boldsymbol{\theta}) = |g(\boldsymbol{\theta}, \mathbf{x}_q) - \mathbf{t}_q| \tag{34}$$

where $g(\boldsymbol{\theta}, \mathbf{x}_q)$ denotes the last layer output of the neural network when $\mathbf{x}_q$ is the input and $\mathbf{t}_q$ is a scalar output. For all classification problems with the exception of GPT-2, we use the measurement function:

$$f_{z_q}(\boldsymbol{\theta}) = -\log \frac{\sigma(g(\boldsymbol{\theta}, \mathbf{x}_q))_{\mathbf{t}_q}}{1 - \sigma(g(\boldsymbol{\theta}, \mathbf{x}_q))_{\mathbf{t}_q}} \tag{35}$$

$$= -g(\boldsymbol{\theta}, \mathbf{x}_q)_{\mathbf{t}_q} + \log \left( \sum_i \exp g(\boldsymbol{\theta}, \mathbf{x}_q)_i - \exp g(\boldsymbol{\theta}, \mathbf{x}_q)_{\mathbf{t}_q} \right), \tag{36}$$

where $\sigma$ denotes the softmax function and the subscript $\mathbf{t}_q$ refers to the taking the entry corresponding to the correct class. This measurement function is identical to the one found in [22]. For GPT-2, we use the training loss as the measurement function.

## F.2 Experiment Details

Table 2 shows the architecture and hyperparameters used to compute the final parameters $\theta^s$, which we use to run EKFAC-IF, EKFAC-SOURCE, ASTRA-IF and ASTRA-SOURCE. The first column shows the size of the training dataset and the number of query examples in each setting. The third column shows the hyperparameters used to train the models in each setting; we use the same hyperparameters as reported in Bae et al. [19]. We estimate the expected ground-truth in the left-hand-side of Equation 10 with an empirical average – the last column in Table 2 shows the number of repeats *per mask* $\mathcal{S}_j$ (i.e., number of $\xi$ sampled) to estimate this value. All settings use a constant learning rate, with the exception of CIFAR-10, which uses a cyclic learning rate schedule.

Table 2: **Summary of training details.**

| Dataset | Architecture | Hyperparameters | Ground-truth Retraining |
|---|---|---|---|
| **UCI Concrete** 896 training examples 103 query examples | MLP - 4 Layers (128, 128, 128) Hidden Units ReLU activation | SGD w/ momentum Learning rate: $3 \times 10^{-2}$ Weight decay: $10^{-5}$ Momentum: 0.9 Batch size: 32 Epochs: 20 | Repeats: 100 Masks: 100 Total: 10,000 |
| **UCI Parkinsons** 5,280 training examples 100 query examples | MLP - 4 Layers (128, 128, 128) Hidden Units ReLU activation | SGD w/ momentum Learning rate: $10^{-2}$ Weight decay: $3 \times 10^{-5}$ Momentum: 0.9 Batch size: 32 Epochs: 20 | Repeats: 100 Masks: 100 Total: 10,000 |
| **MNIST (Subset)** 6,144 training examples 100 query examples | MLP - 4 Layers (512, 256, 128) Hidden Units ReLU activation | SGD w/ momentum Learning rate: $3 \times 10^{-2}$ Weight decay: $10^{-3}$ Momentum: 0.9 Batch size: 64 Epochs: 20 | Repeats: 50 Masks: 100 Total: 5,000 |
| **FashionMNIST (Subset)** 6,144 training examples 100 query examples | MLP - 4 Layers (512, 256, 128) Hidden Units ReLU activation | SGD w/ momentum Learning rate: $3 \times 10^{-2}$ Weight decay: $10^{-3}$ Momentum: 0.9 Batch size: 64 Epochs: 20 | Repeats: 50 Masks: 100 Total: 5,000 |
| **CIFAR-10 (Subset)** 3,072 training examples 100 query examples | ResNet-9 [78] | SGD w/ momentum Peak learning rate: 0.4 Weight decay: $10^{-3}$ Momentum: 0.9 Batch size: 512 Epochs: 25 | Repeats: 50 Masks: 100 Total: 5,000 |
| **WikiText-2** 4,656 training sequences 512 sequence length 100 query sequences | GPT-2 [81] | AdamW Learning rate: $3 \times 10^{-5}$ Weight decay: $10^{-2}$ Batch size: 8 Epochs: 3 | Repeats: 10 Masks: 100 Total: 1,000 |
| **FashionMNIST-N** 6,144 training examples 100 query examples 30% of the dataset randomly relabeled | MLP - 4 Layers (512, 256, 128) Hidden Units ReLU activation | SGD w/ momentum Learning rate: $10^{-2}$ Weight decay: $3 \times 10^{-5}$ Momentum: 0.9 Batch size: 64 Epochs: 3 | Repeats: 50 Masks: 100 Total: 5,000 |

Table 3 shows the details for the various TDA algorithms we use in our experiments.

For EKFAC-IF, we use the same damping as the corresponding ASTRA-IF for comparability. SOURCE provides a natural value for damping from its derivation: $\lambda_\ell = 1/\overline{\eta}_\ell K_\ell$ [19], allowing us to sidestep tuning the damping term. For both EKFAC-IF and ASTRA-IF, we use the damping value implied by SOURCE for comparability between influence functions and SOURCE: we take the average $\lambda_\ell$ implied by SOURCE for each segment and weigh them by the total iterations $K_\ell$ in segment $\ell$ as our influence functions damping value.

EKFAC-IF, EKFAC-SOURCE, ASTRA-IF, and ASTRA-SOURCE all compute influence on the same set of layers. For MLP architectures, we compute influence on all layers. For ResNet-9, we

compute influence on MLP and convolution layers. For GPT-2, we compute influence only on MLP layers in line with Grosse et al. [7].

Table 3: **Summary of TDA Algorithm Details**

| Dataset | ASTRA-IF Details | ASTRA-SOURCE Details |
|---|---|---|
| **UCI Concrete** | We run ASTRA for 200 iterations, and use GGN damping factor $\lambda = 0.0017$, preconditioner damping factor $\tilde{\lambda} = 0.0017$, learning rate $\alpha = 0.1\lambda$, batch-size 256, SGD w/ 0.9 momentum, and decay the learning rate by 0.5 every 50 iterations. | We use 3 segments. For each segment, we run ASTRA for 200 iterations, and use preconditioner damping factor equal to the GGN damping factor $\tilde{\lambda}_\ell = \lambda_\ell = 1/\overline{\eta}_\ell K_\ell$, learning rate $\alpha_\ell = 0.1\lambda_\ell$, batch-size 256, SGD w/ 0.9 momentum, and decay the learning rate by 0.5 every 50 iterations. |
| **UCI Parkinsons** | We run ASTRA for 200 iterations, and use GGN damping factor $\lambda = 0.00091$, preconditioner damping factor $\tilde{\lambda} = 0.00091$, learning rate $\alpha = 0.1\lambda$, batch-size 256, SGD w/ 0.9 momentum, and decay the learning rate by 0.5 every 50 iterations. | We use 3 segments. For each segment, we run ASTRA for 200 iterations, and use preconditioner damping factor equal to the GGN damping factor $\tilde{\lambda}_\ell = \lambda_\ell = 1/\overline{\eta}_\ell K_\ell$, learning rate $\alpha_\ell = 0.1\lambda_\ell$, batch-size 256, SGD w/ 0.9 momentum, and decay the learning rate by 0.5 every 50 iterations. |
| **MNIST** | We run ASTRA for 200 iterations, and use GGN damping factor $\lambda = 0.0052$, preconditioner damping factor $\tilde{\lambda} = 0.0052$, learning rate $\alpha = 0.1\lambda$, batch-size 256, SGD w/ 0.9 momentum, and decay the learning rate by 0.5 every 50 iterations. | We use 3 segments. For each segment, we run ASTRA for 200 iterations, and use preconditioner damping factor equal to the GGN damping factor $\tilde{\lambda}_\ell = \lambda_\ell = 1/\overline{\eta}_\ell K_\ell$, learning rate $\alpha_\ell = 0.1\lambda_\ell$, batch-size 256, SGD w/ 0.9 momentum, and decay the learning rate by 0.5 every 50 iterations. |
| **FashionMNIST** | We run ASTRA for 200 iterations, and use GGN damping factor $\lambda = 0.0052$, preconditioner damping factor $\tilde{\lambda} = 0.0052$, learning rate $\alpha = 0.1\lambda$, batch-size 256, SGD w/ 0.9 momentum, and decay the learning rate by 0.5 every 50 iterations. | We use 3 segments. For each segment, we run ASTRA for 200 iterations, and use preconditioner damping factor equal to the GGN damping factor $\tilde{\lambda}_\ell = \lambda_\ell = 1/\overline{\eta}_\ell K_\ell$, learning rate $\alpha_\ell = 0.1\lambda_\ell$, batch-size 256, SGD w/ 0.9 momentum, and decay the learning rate by 0.5 every 50 iterations. |
| **CIFAR-10** | We run ASTRA for 200 iterations, and use GGN damping factor $\lambda = 0.014$, preconditioner damping factor $\tilde{\lambda} = 0.014$, learning rate $\alpha = 0.01\lambda$, batch-size 128, SGD w/ 0.9 momentum, and decay the learning rate by 0.5 every 50 iterations. | We use 3 segments. For each segment, we run ASTRA for 200 iterations, and use preconditioner damping factor equal to the GGN damping factor $\tilde{\lambda}_\ell = \lambda_\ell = 1/\overline{\eta}_\ell K_\ell$, learning rate $\alpha_\ell = 0.01\lambda_\ell$, batch-size 128, SGD w/ 0.9 momentum, and decay the learning rate by 0.5 every 50 iterations. |
| **WikiText-2** | We run ASTRA for 300 iterations, and use GGN damping factor $\lambda = 0.011$, preconditioner damping factor $\tilde{\lambda} = 0.011$, learning rate $\alpha = \lambda$, batch-size 16, SGD w/ 0.9 momentum, and decay the learning rate by 0.9 every 100 iterations. | We use 3 segments. For each segment, we run ASTRA for 300 iterations, and use preconditioner damping factor equal to the GGN damping factor $\tilde{\lambda}_\ell = \lambda_\ell = 1/\overline{\eta}_\ell K_\ell$, learning rate $\alpha_\ell = \lambda_\ell$, batch-size 8, SGD w/ 0.9 momentum, and decay the learning rate by 0.9 every 100 iterations. |
| **FashionMNIST-N** | We run ASTRA for 200 iterations, and use GGN damping factor $\lambda = 0.10$, preconditioner damping factor $\tilde{\lambda} = 0.10$, learning rate $\alpha = 0.1\lambda$, batch-size 256, SGD w/ 0.9 momentum, and decay the learning rate by 0.5 every 50 iterations. | We use 3 segments. For each segment, we run ASTRA for 200 iterations, and use preconditioner damping factor equal to the GGN damping factor $\tilde{\lambda}_\ell = \lambda_\ell = 1/\overline{\eta}_\ell K_\ell$, learning rate $\alpha_\ell = 0.1\lambda_\ell$, batch-size 256, SGD w/ 0.9 momentum, and decay the learning rate by 0.5 every 50 iterations. |

**Other Baselines** In Figure 2, in addition to the EKFAC-based baselines, we also compare ASTRA against TracIn [83] and TRAK [22]. We use the same checkpoints for TracIn as SOURCE for comparability. TRAK applies random projections to reduce the computation and memory footprint – for MNIST, FashionMNIST(-N) and UCI Parkinsons, we use projection dimensions of 4,096 and for UCI Concrete, we use projection dimension of 512, consistent with [19]. For CIFAR-10, we do a hyperparameter sweep among [128, 256, 512, 1024, 2048, 4096, 8192, 20480] for the best hyperparameter (512). We omit TRAK's results on GPT-2 due to lack of publicly available implementations.

**Training Curve Details** In Figure 3, we compare ASTRA-IF and SNI on an arbitrary $z_q$, using 10 seeds and report mean values along with shaded regions representing 1 standard error. To compute both ASTRA-IF and the baseline SNI in Figure 3, we conduct a hyperparameter sweep for the learning rate from $10^0$ to $10^{-5}$ in increments of one order of magnitude with momentum set to zero, and use the best learning rate based on the average training objective over the last ten iterations. Both SNI and ASTRA-IF use the same damping values and batch sizes for Figure 3, as listed in Table 3 for comparability. The learning rates used were $10^{-2}$ for all settings for ASTRA-IF, and the best learning rates for SNI were $1, 0.1, 0.01, 0.1$ for MNIST, FashionMNIST, CIFAR-10, and UCI Concrete respectively.

**Eigendecomposition Details** In Figure 4, we compare the performance of influence functions computed with ASTRA and SNI after projecting to various subspaces. This experiment uses a smaller damping of $10^{-4}$ for both MNIST and FashionMNIST to be able to discern the impact of directions of low curvature on influence functions performance. We use the same batch sizes as disclosed in Table 3 and no momentum. For ASTRA-IF, we used a learning rate of $10^{-4}$ for both settings. For the baseline SNI, the learning rates were 0.1 and 0.01 respectively for MNIST and FashionMNIST, which we found to give the strongest LDS for the subspace corresponding to $S_5$.

**Compute Resources** A shared cluster was used (both internal and external), consisting of a mix of A6000 (48GB), A100 (80GB) and H100 (80GB) GPUs, which were used to conduct all experiments. Computing the ground truth of the LDS experiments in Figure 1 is an expensive component of the overall compute to replicate the experiments. For the most expensive setting, fine-tuning GPT-2 with WikiText-2, computing ground-truth costs at most 500 hours of compute with the resources listed above. Similarly, running EKFAC-IF, EKFAC-SOURCE, ASTRA-IF and ASTRA-SOURCE for the GPT-2 setting is the most expensive of all the settings. Running ASTRA-SOURCE for the GPT-2 setting costs at most 40 hours with the compute resources listed above.

**Statistical Significance** For LDS experiments, we provide error bars indicating 1 standard error for all one-model predictions estimated with 10 seeds. For the training curve (Figure 3) and the eigendecomposition experiments (Figure 4), we show 1 standard error estimated with 10 seeds. The standard error is computed by taking the standard deviation $s$ of the computed metric, divided by $\sqrt{10}$.

**Assets** We use `pytorch` 2.5.0 and the publicly available `kronfluence` package for our experiments, which can be found at https://github.com/pomonam/kronfluence.

# G  Implications of Truncated Neumann Series

In Section 2.1, we introduced the connection between NI and the iHVP approximation. Iterative iHVP approximation algorithms like LiSSA [23] usually requires a large number of iterations to achieve good iHVP approximations [7, 17]. In practice, the number of iterations in the algorithm is usually set with the assumption that the approximation converges within the iterations (e.g., 5000 in [17]). While a large number of iterations makes convergence likely, it is not a guarantee, and raising the number of iterations implies additional computational cost. We demonstrate that using fewer iterations and stopping before convergence can be interpreted as adding an additional damping term to the iHVP approximation. There is existing work noting that the truncation of Neumann series can be viewed as increased damping [29], but here we present it with the derivation technique found in

[19]. We derive the following expression for the truncated Neumann series with $J$ iterations:

$$\alpha \sum_{j=0}^{J-1} (\mathbf{I} - \alpha\mathbf{G} - \alpha\lambda\mathbf{I})^j = (\mathbf{G} + \lambda\mathbf{I})^{-1}(\mathbf{I} - (\mathbf{I} - \alpha\mathbf{G} - \alpha\lambda\mathbf{I})^J) \tag{37}$$

$$\approx (\mathbf{G} + \lambda\mathbf{I})^{-1}(\mathbf{I} - \exp(-\alpha J\mathbf{G} - \alpha J\lambda\mathbf{I})) \tag{38}$$

$$\approx (\mathbf{G} + \hat{\lambda}\mathbf{I})^{-1}, \tag{39}$$

where $\hat{\lambda} = \lambda + \alpha^{-1}J^{-1}$. Equation 37 and Equation 38 utilize the finite series and the matrix exponential definition, respectively. Given that the matrices in Equation 38 commute, we can express it as the following matrix function:

$$F(\sigma) := \frac{1 - \exp(-\alpha J\sigma - \alpha J\lambda)}{\sigma + \lambda}. \tag{40}$$

Let $\mathbf{G} = \mathbf{Q}\mathbf{D}\mathbf{Q}^\top$ be the eigendecomposition, where $\sigma_j$ denotes the $j$th eigenvalue of $\mathbf{G}$. The expression in Equation 38 can be interpreted as applying the function $F(\sigma)$ to each eigenvalue $\sigma$ of $\mathbf{G}$. In high-curvature directions, this term asymptotically approaches $1/\sigma + \lambda$, whereas in low-curvature directions with small values of $\lambda$, it tends toward $\alpha J$.

The qualitative behavior of the function $F$ can be captured by $F_{\text{inv}}$, defined as:

$$F_{\text{inv}}(\sigma) := \frac{1}{\sigma + \lambda + \alpha^{-1}J^{-1}} \tag{41}$$

Applying $F_{\text{inv}}$ to the Hessian results in approximating Equation 39 with a modified damping term $\hat{\lambda} := \lambda + 1/\alpha J$. Hence, the truncated version can be interpreted as incorporating a larger damping term, by $1/\alpha J$. The implicitly larger damping term affects the iterative updates and adds additional difficulty to tuning the hyperparameters of iterative iHVP solvers. In ASTRA, we leverage the EKFAC approximation as a preconditioner for the iHVP approximation to improve the conditioning of the problem, which mitigates this issue.

# H   Limitations & Broader Impact

**Limitations**   We have addressed the problem of computing accurate iHVPs for TDA. As we outlined in Section 2.2, in addition to the computing the iHVP, a large component of the cost in computing influence functions for both EKFAC-IF and ASTRA-IF when scanning the dataset for highly influential examples is taking the dot product with all the training example gradients, which is an orthogonal but important issue that we do not address. We also noted in our experiments that in some cases, the bottleneck for influence function performance in TDA is not an inaccurate iHVP, but other factors such as violating the fundamental assumptions involved in the influence functions derivation. In these cases, the benefits from an improved iHVP approximation may not materialize until other bottlenecks are resolved. For example, the FashionMNIST experiments show a large increase in performance after ensembling relative to the 1-model scores obtained by an improved iHVP approximation, which suggests that stochasticity in the training procedure may be the dominant factor hindering TDA performance. Finally, for ASTRA-SOURCE, we present a way in which we can apply the iHVP computation, but we leave to future work to explore various ways to compute the average GGN $\overline{\mathbf{G}}_\ell$, which may further improve performance.

**Broader impact**   Our work improves the accuracy of the iHVP approximation for use in TDA. The algorithmic improvements we present do not have direct societal impact. However, improved TDA can provide insights into the relation between training data and model behavior. From this perspective, our work has similar broader impact to other work in TDA, sharing similar potential benefits – namely, enhanced interpretability, transparency, and fairness in AI systems and share similar risks. Specifically, advancing TDA can help us understand models through the lens of training data, which can be applied to many domains such as building more equitable and transparent machine learning systems [12, 13], and investigating questions of intellectual property and copyright [11, 14, 15]. On the flip side, TDA can also be used to maliciously to craft data poisoning attacks and could result in models with undesirable behavior. It is important that TDA algorithms are improved with mitigation of the risks.

