# OpenReview forum: "Better Training Data Attribution via Better Inverse Hessian-Vector Products"
_NeurIPS.cc/2025/Conference — NeurIPS 2025 poster_

### Official Review · Reviewer_ZrKT · 2025-06-27

**Clarity:** 2
**Significance:** 2
**Originality:** 3
**Rating:** 4
**Confidence:** 3

**Summary:**

This paper focuses on improving the accuracy and reliability of influence function-based training data attribution (TDA) methods, particularly in architectures involving convolutions, where existing EKFAC-based approaches tend to underperform. The authors propose ASTRA, a new algorithm that leverages EKFAC as a preconditioner for Stochastic Neumann Iteration (SNI) to compute inverse Hessian-vector products (iHVPs) more accurately and efficiently. By addressing the computational and numerical challenges in iHVP approximation, ASTRA enhances the quality of influence estimation.

**Questions:**

1. While ASTRA is empirically shown to be effective, does the method come with any theoretical guarantees or bounds on convergence or approximation error, especially compared to standard Neumann series iteration?

2. EKFAC and second-order approximations can be computationally demanding. Could the authors comment on how ASTRA scales to large-scale architectures, such as modern transformers or deep convolutional networks? Are there specific memory or runtime trade-offs?

3. The paper claims improved performance in convolutional settings. Are there particular architectural patterns (e.g., depth, residual connections, attention) where ASTRA struggles or underperforms? Additional results or qualitative insights would be helpful.

**Ethical Concerns:**

["NO or VERY MINOR ethics concerns only"]

**Final Justification:**

After reading the rebuttal, I find that the authors have provided some reasonable responses, and they appear to have a clear understanding of both the strengths and the limitations of their method. While I am not deeply familiar with the full body of literature in this specific subfield and have not reviewed all the cited works, my evaluation is based on general research experience and standard expectations in the field. It should be viewed as a perspective from outside the core community, and I hope it offers a constructive and useful reference for the judgment of other reviewers.

**Limitations:**

I do not see any potential negative societal impact.

**Paper Formatting Concerns:**

I did not notice any major formatting issues in the paper.

**Quality:**

3

**Strengths And Weaknesses:**

Strengths:

1. The paper proposes ASTRA, a well-motivated algorithm that uses EKFAC as a preconditioner for Stochastic Neumann Iteration (SNI), offering a more accurate and cost-effective way to approximate inverse Hessian-vector products (iHVPs), which are central to influence-function-based TDA.

2. While prior work has questioned the practical value of influence functions due to poor iHVP approximations, this work shows that with improved estimation (via ASTRA), influence scores can regain predictive utility, even in settings that violate the original theoretical assumptions.

3. The paper identifies and targets a key weakness in existing EKFAC-based methods—namely their poor performance on convolutional networks—and proposes a method that generalizes better to such architectures.

Weaknesses:

1. Although ASTRA is more efficient than full Hessian inversion, it still requires access to second-order statistics and matrix-vector products. The paper does not fully explore how well ASTRA scales to large-scale models (e.g. ResNet-152 on Imagenet), where such operations may be prohibitively expensive.

---

> ### Author Rebuttal · Authors · 2025-07-30
>
> Thank you for your positive review. We appreciate your comments about the algorithm being “well-motivated”. We are also glad you appreciate the unique improvement ASTRA provides for convolution architectures. We will address your questions and concerns below line by line:
>
> > it still requires access to second-order statistics and matrix-vector products
>
> The reason for second order statistics being necessary is that the classic influence functions derivation contains the Hessian [1]. Past work has shown that inclusion of second-order information significantly improves efficacy of TDA — we can compare TDA outcomes measured via LDS for gradient dot products with methods that use second order information: (e.g. EKFAC influence and ASTRA influence), and in general we observe that better second order approximation results in a better TDA outcome. Both our results and past results in [2, 3, 4] show that this is the case.
>
> You are correct in implying that incorporating second order information increases the computational burden. However, strong TDA methods such as EKFAC influence already computes second order information. In our paper, we show that at a limited additional cost, by repurposing the EKFAC approximation to the Hessian as a preconditioner and preconditioning the SNI update, we can achieve better TDA outcomes.
>
> > The paper does not fully explore how well ASTRA scales to large-scale models (e.g. ResNet-152 on Imagenet)
>
> Evaluation for TDA is very expensive, since the very purpose of TDA algorithms is to substitute for repeated retraining, which is the ground truth we measure against. Thus, the scale of our experiments is not bottlenecked by our TDA method, but rather by our ability to compute ground truth to compute LDS. In our results in Figure 2, each setting required us to train a model over 1,000 times to obtain a stable estimate of ground truth.
>
> **New Experiment 1:**
>
> However, we ran a larger experiment involving LDS, in which we trained a subset of CIFAR-100 on ResNet18. The LDS scores are:
>
> EKFAC IF 1 model, EKFAC IF 10 model: 0.143, 0.244
>
> ASTRA IF 1 model, ASTRA IF 10 model: 0.232, 0.478
>
> Again, our method ASTRA-IF provides a significant boost compared to EKFAC-IF.
>
> **New Experiment 2:**
>
> Despite not being able to estimate ground truth for large models to compute LDS, we reran our method ASTRA-IF on a 1.5B parameter model GPT2-XL to see how the loss in Equation 4 behaves compared to influence computed with SNI. While we cannot paste the plots here due to NeurIPS rebuttal rules, we will describe the result:
>
> Starting from the same initialization, so that the loss at iteration 0 are the same for both methods, we did a learning rate sweep for both methods, increasing the learning rate until both methods diverged. We chose the best learning rates at a fixed iteration for each method and plotted the values of Equation 4 for both our method ASTRA-IF and SNI. The loss curve as defined in Equation 4 is substantially lower for our method compared to SNI for the same number of iterations: the best SNI result achieved a loss of approximately $-10^8$ , while EKFAC achieved $-5 \\times 10^8$ after 40 iterations. The loss of our method was lower than SNI throughout the entire 40 iterations despite starting out the same at iteration 0.
>
> > does the method come with any theoretical guarantees or bounds on convergence or approximation error
>
> Our algorithm is guaranteed to converge for a sufficiently small learning rate, an appropriate learning rate schedule, and the choice of $\\mathbf{P} := \\mathbf{G}_{\\text{EKFAC}}$ as is done in our paper.
>
> To see this, note that our update rule in Equation 5 is the mini-batch gradient descent rule for the quadratic:
>
> $$\\frac{1}{2} \\boldsymbol{\\theta}^\\top (\\mathbf{P} + \\tilde{\\lambda} \\mathbf{I})^{-1}(\\mathbf{G}+ \\lambda \\mathbf{I})\\boldsymbol{\\theta} - \\boldsymbol{\\theta}^\\top (\\mathbf{P} + \\tilde{\\lambda} \\mathbf{I})^{-1}\\nabla_{\\boldsymbol{\\theta}} f_{z_q} (\\boldsymbol{\\theta}^s)$$
>
> and for our choice of $\\mathbf{P} := \\mathbf{G}_{\\text{EKFAC}}$, $(\\mathbf{P} + \\tilde{\\lambda} \\mathbf{I})^{-1}(\\mathbf{G}+\\lambda \\mathbf{I})$ is positive definite. We thus inherit properties of mini-batch gradient descent on strongly convex quadratics.
>
> However, we cannot make any theoretical claims about the converge rate, since the condition number after preconditioning depends on the quality of the approximation $\\mathbf{G}_{\\text{EKFAC}}$, which does not carry theoretical guarantees, but is empirically known to work well [5, 6].
>
> > Could the authors comment on how ASTRA scales to large-scale architectures, such as modern transformers or deep convolutional networks. Are there specific memory or runtime trade-offs?
>
> Compared with SNI, our method ASTRA requires an additional matrix vector product to compute the product of $(\\mathbf{P}+ \\lambda \\mathbf{I})^{-1}$ with $(\\tilde{\\mathbf{G}}_k+ \\lambda \\mathbf{I})\\boldsymbol{\\theta}_k$, whose computation costs slightly more than one backward pass. In terms of memory, compared with SNI, we need to additionally keep the EKFAC statistics in memory, and the exact memory cost of EKFAC depends on the model architecture. We have a simplified illustration on line 113 in which the memory cost scales linearly with the dimension of the network. This is a large reduction compared to storing the Hessian naively, which scales quadratically with the dimension of the network.
>
> > Are there particular architectural patterns (e.g., depth, residual connections, attention) where ASTRA struggles or underperforms?
>
> For influence functions, typically a decision is made as to which layers/modules we compute influence over. In our paper, we compute influence only over fully connected and convolution layers. For LLMs, we only consider influence of the fully connected layers, in line with [4]. We explain this in lines 1244-1247 in the Appendix. The experiments involving GPT2 and ResNet include a residual connection. However, we do not observe any reliable pattern with respect to depth and residual connections.
>
>
> ```
> References
> [1]: Koh, Pang Wei et al. “Understanding Black-box Predictions via Influence Functions”.
> [2]: Bae, Juhan et al. “Training Data Attribution via Approximate Unrolled Differentiation”.
> [3]: Choe, Sang Keun et al. “What is Your Data Worth to GPT? LLM-Scale Data Valuation with Influence Functions”.
> [4]: Grosse, Roger et al. “Studying Large Language Model Generalization with Influence Functions”.
> [5]: Martens, James et al. “Optimizing Neural Networks with Kronecker-factored Approximate Curvature”.
> [6]: George, Thomas et. al. "Fast Approximate Natural Gradient Descent in a Kronecker-factored Eigenbasis".
> ```

---

### Official Review · Reviewer_LxJj · 2025-07-02

**Clarity:** 3
**Significance:** 3
**Originality:** 3
**Rating:** 4
**Confidence:** 2

**Summary:**

The paper introduces ASTRA, an algorithm that uses EKFAC-preconditioner on Neumann series iterations, to arrive at iHVP approximation for training data attribution more accurately and efficiently. On tabular, vision, and laguage tasks, ASTRA speeds up iHVP approximation and beats baseline methods on accuracy. The paper also provides an exploration on how curvature directions affect influence functions performance.

**Questions:**

How would the method scale with larger models (billion parameters)? If empirical analysis cannot be done due to compute constraint, is there estimate / small-scale scaling behavior experiment that can be done?

**Ethical Concerns:**

["NO or VERY MINOR ethics concerns only"]

**Final Justification:**

Author's rebuttal addresses some of my concerns. I decide to maintain my score since points I raised are still valid.

**Limitations:**

Yes.

**Quality:**

3

**Strengths And Weaknesses:**

Strength:
- Extensive evaluation on a lot of tasks, modality, and architecture. The authors do not restrict themselves to one data modality.  They benchmark on small regression MLPs, classic image MLPs, a convolutional ResNet-9, and an autoregressive GPT-2 fine-tune.
- Rigorous mathematical derivation. Section 3 exposes derivation step by step clearly.
- Beside the method itself, the paper also provides a further exploration of low curvature impacts that underpins the paper's hypothesis. This analysis not only validates the method but also gives practitioners guidance on where to focus optimisation effort.


Weakness:
- Lacks discussion on how the method would scale with larger models. The introduction acknowledges that EKFAC can scale to billion-parameter language models, yet the largest empirical target is a 124 M-parameter GPT-2 fine-tune. Modern language and vision models are usually billion parameter levels. It's unclear how the method would scale to larger models.
- For some evaluations, performance depends a lot on ensemble. This weakens the out-of-the-box appeal.

---

> ### Author Rebuttal · Authors · 2025-07-30
>
> Thank you for your positive review. We appreciate that you found our evaluation “extensive”, and our mathmatical derivation “rigorous”. In particular, we are glad that you found that Section 6 (which shows that low curvature directions really matter for TDA) useful for practitioners. We believe it is quite important that practitioners understand that directions of low curvature carry a lot of predictive information for TDA. We now address your questions and concerns line by line:
>
> > Lacks discussion on how the method would scale with larger models
>
> Evaluation for TDA is very expensive, since the very purpose of TDA algorithms is to substitute for repeated retraining, which is the ground truth we measure against. Thus, the scale of our experiments is not bottlenecked by our TDA method, but rather by our ability to compute ground truth. In our main results, each setting required us to train a model over 1,000 times to obtain a stable estimate of ground truth.
>
> **New Experiment 1:**
>
> However, we ran a larger experiment involving LDS, in which we trained CIFAR-100 on ResNet18. The LDS scores are:
>
> EKFAC IF 1 model, EKFAC IF 10 model: 0.143, 0.244
>
> ASTRA IF 1 model, ASTRA IF 10 model: 0.232, 0.478
>
> Again, our method ASTRA-IF provides a significant boost compared to EKFAC-IF.
>
> **New Experiment 2:**
> Despite not being able to estimate ground truth for large models to compute LDS, we reran our method ASTRA-IF on a 1.5B parameter model GPT2-XL to see how the loss in Equation 4 behaves compared to influence computed with SNI. While we cannot paste the plots here due to NeurIPS rebuttal rules, we will describe the result:
>
> Starting from the same initialization, so that the loss at iteration 0 are the same, we did a learning rate sweep for both methods, increasing the learning rate until both methods diverged. We chose the best learning rates at a fixed iteration for each method and plotted the values of Equation 4 for both our method ASTRA-IF and SNI. The loss curve as defined in Equation 4 is substantially lower for our method compared to SNI for the same number of iterations: the best SNI result achieved a loss of approximately $-10^8$ , while EKFAC achieved $-5 \\times 10^8$ after 40 iterations. The loss of EKFAC preconditioning was lower than SNI throughout the entire 40 iterations despite starting out the same at iteration 0.
>
> Finally, we can provide some color on the costs of our algorithm. Recall that our algorithm optimizes Equation 4 using preconditioned SNI. Compared with regular SNI, our method ASTRA requires an additional matrix vector product to compute the product of $(\\mathbf{P}+ \\lambda \\mathbf{I})^{-1}$ with $(\\tilde{\\mathbf{G}}_k+ \\lambda \\mathbf{I})\\boldsymbol{\\theta}_k$, whose computation costs slightly more than one backward pass. In terms of memory, compared with SNI, we need to additionally keep the EKFAC statistics in memory, and the exact memory cost of EKFAC depends on model architecture. We have a simplified illustration on line 113 in which the memory cost scales linearly with the dimension of the network. This is a large reduction compared to storing the Hessian naively, which scales quadratically with the dimension of the network.
>
>
> > For some evaluations, performance depends a lot on ensemble. This weakens the out-of-the-box appeal.
>
> Figure 2 compares 1 model performance, as well as 10 model ensembled performance. Relative to EKFAC-baselines, in both cases, we find improvements using our method.

---

> > ### Comment · Reviewer_LxJj · 2025-08-07
> >
> > Thanks authors for the detailed response and additional experiments. They addressed my concerns. I will maintain my rating.

---

### Official Review · Reviewer_CuEj · 2025-07-03

**Clarity:** 2
**Significance:** 4
**Originality:** 4
**Rating:** 5
**Confidence:** 3

**Summary:**

The paper introduces ASTRA, a solver that pre-conditions Stochastic Neumann Iterations (SNI) with the cached eigenbasis from Eigenvalue-Corrected K-FAC (EKFAC). One EKFAC step supplies an inexpensive initial iHVP; a few hundred extra Neumann steps refine it toward the unbiased solution. On seven benchmarks—MLPs, ResNet-9, and GPT-2 fine-tuning—ASTRA consistently raises the Linear Data-modeling Score (LDS), e.g. CIFAR-10 jumps from 0.25→0.60, while adding <10 % runtime over EKFAC. Eigen-space analysis shows that rapid progress in low-curvature directions underlies these gains.

**Questions:**

- How does ASTRA scale when EKFAC eigenupdates dominate cost on LLMs? Can ASTRA refine low-rank curvature methods (e.g. LoGRA) to combine memory savings with higher fidelity?

**Ethical Concerns:**

["NO or VERY MINOR ethics concerns only"]

**Quality:**

4

**Strengths And Weaknesses:**

**Strengths**
- Novelty: Simple yet novel fusion of EKFAC and SNI that had not been explored for TDA.
- Principled: Clear derivations (truncation ⇔ implicit damping) and principled pre-conditioning.
- Convincing experiments: Seven datasets, timing curves, eigen-space breakdowns, and ensemble analysis.
- Practical impact: Drop-in replacement for existing influence-function and unrolled-differentiation pipelines with minimal compute overhead.
- Insightful analysis: Demonstrates that low-curvature directions dominate influence quality, guiding future work.

**Weaknesses**
- Dense writing: Readers unfamiliar with EKFAC or SNI may struggle; a brief background / preliminaries section would help.
- Limited model scale: The largest test model is GPT-2 (124 M params); behavior on larger-scale vision and language models unclear.
- Baseline coverage: No empirical comparison to recent TDA methods such as LoGRA, TRAK, or MAGIC.

---

> ### Author Rebuttal · Authors · 2025-07-30
>
> Thank you for your positive review. Based on your comments, we are very happy that you appreciate the subtleties in our work and we notably appreciate the time you put into reading our paper carefully. In particular, we are happy that you find the method “novel”,  the analysis “insightful”, the derivations “clear”, and the experiments “convincing”.  We now address your questions and concerns line by line.
>
> > Dense writing: Readers unfamiliar with EKFAC or SNI may struggle; a brief background / preliminaries section would help.
>
> Thank you for this feedback. There are extended preliminaries which cover this background material in Appendix B. We are happy to move some of the preliminaries to the body to appeal to those without a background in EKFAC or SNI.
>
> > Limited model scale: The largest test model is GPT-2 (124 M params); behavior on larger-scale vision and language models unclear.
> >
>
> Evaluation for TDA is very expensive, since the very purpose of TDA algorithms is to substitute for repeated retraining, which is the ground-truth we measure against. Thus, the scale of our experiments is not bottlenecked by our TDA method, but rather by our ability to evaluate the outcome. In our main results, each setting required us to train a model over 1,000 times.
>
> **New Experiment 1:**
>
> However, we ran a larger experiment involving LDS, in which we trained a subset of CIFAR-100 on ResNet18. The LDS scores are:
>
> EKFAC IF 1 model, EKFAC IF 10 model: 0.143, 0.244
>
> ASTRA IF 1 model, ASTRA IF 10 model: 0.232, 0.478
>
> Again, our method ASTRA-IF provides a significant boost compared to EKFAC-IF.
>
> **New Experiment 2:**
>
> Despite not being able to estimate ground truth for large models to compute LDS, we also reran our method ASTRA-IF on a 1.5B parameter model GPT2-XL to see how the loss in Equation 4 behaves compared to influence computed with SNI. While we cannot paste the plots here due to NeurIPS rebuttal rules, we will describe the result:
>
> Starting from the same initialization, so that the loss at iteration 0 are the same, we did a learning rate sweep for both methods, increasing the learning rate until both methods diverged. We chose the best learning rates at a fixed iteration for each method and plotted the values of Equation 4 for both our method ASTRA-IF and SNI. The loss curve as defined in Equation 4 is substantially lower for our method compared to SNI for the same number of iterations: the best SNI result achieved a loss of approximately $-10^8$ , while EKFAC achieved $-5 \\times 10^8$ after 40 iterations. The loss of EKFAC preconditioning was lower than SNI throughout the entire 40 iterations despite starting out the same at iteration 0.
>
> > Baseline coverage: No empirical comparison to recent TDA methods such as LoGRA, TRAK, or MAGIC.
>
> We have added comparisons to TracIn [2] and TRAK [3] to our existing experiments for completeness. Since we are not allowed to add plots, we describe the results here:
>
> **New Baseline 1**:
>
> We ran TracIn for all settings. 1 model and 10 model ensemble TracIn scored below 0.3 LDS for all settings except FashionMNIST-N. TracIn's LDS scores are substantially lower than ASTRA-IF.
>
> We list the 1 model and 10 model scores for TracIn as a tuple (1 model LDS, 10 model LDS), in the following order: MNIST, FashionMNIST, CIFAR-10, UCI Concrete, UCI Parkinsons, WikiText2, FashionMNIST-N.
> (0.154, 0.191), (0.201, 0.223), (0.107, 0.121), (0.191, 0.227), (0.151, 0.165), (0.1580, 0.1587), (0.577, 0.586).
>
> Note: TracIn uses the same checkpoints as ASTRA-SOURCE for comparability.
>
> **New Baseline 2**:
>
> We also ran TRAK for all settings except GPT-2, since there does not exist a public implementation for language modeling tasks. Both 1 model and 10 model ensemble TRAK score below 0.4 for all settings we ran.
>
> We list the 1 model and 10 model scores for TRAK as a tuple (1 model LDS, 10 model LDS), in the following order: MNIST, FashionMNIST, CIFAR-10, UCI Concrete, UCI Parkinsons, FashionMNIST-N.
> (0.132, 0.340), (0.102, 0.246), (0.145, 0.342), (0.218, 0.392), (0.073, 0.194), (0.087, 0.213).
>
> Note: TRAK uses the same projection dimensions as in [4] except CIFAR-10, in which we conduct a hyperparameter sweep among [128, 256, 512, 1024, 2048, 4096, 8192, 20480] and use the best dimension (512).
>
>
> **LoGRA discussion**
> We would *expect* LoGRA to underperform EKFAC influence, since LoGRA computes EKFAC in lower dimensions either after 1) random projections or 2) projection to the PCA eigenbasis using a Kronecker structured projection matrix. In their paper [1], LoGRA underperforms EKFAC, with the benefit of reducing the compute and memory footprint.
>
> **MAGIC discussion**
> We opted to not compare against MAGIC since its publication occurred well past the cutoff of March 1st, 2025 — the NeurIPS 2025 cutoff for contemperaneous work, and we were not able to find a public implementation to use.
>
> > How does ASTRA scale when EKFAC eigenupdates dominate cost on LLMs? Can ASTRA refine low-rank curvature methods (e.g. LoGRA) to combine memory savings with higher fidelity?
>
> In Appendix B, lines 1100-1106, we lay out the main components of computing EKFAC for influence functions, which can give the reader a sense as to how EKFAC scales. But we think your idea here is great — in principle, we can do all our computation in a projected space, which may give us an even more favorable compute vs. accuracy tradeoff, and we think this is a promising direction for future work.
>
> ```
> References:
> [1]: Choe, Sang Keun et al. “What is Your Data Worth to GPT? LLM-Scale Data Valuation with Influence Functions”.
> [2]: Pruthi, Garima et. al. "Estimating Training Data Influence by Tracing Gradient Descent".
> [3]: Park, Sung Min et al. “TRAK: Attributing Model Behavior at Scale".
> [4]: Bae, Juhan et al. “Training Data Attribution via Approximate Unrolled Differentiation”.
> ```

---

> > ### Comment · Reviewer_CuEj · 2025-08-04
> >
> > Thank you for adding these baselines!

---

### Official Review · Reviewer_Kz7b · 2025-07-08

**Clarity:** 1
**Significance:** 2
**Originality:** 4
**Rating:** 4
**Confidence:** 3

**Summary:**

Influence functions and unrolled differentiation involve a computation that resembles an inverse Hessian-vector product (iHVP), which is difficult to numerically compute efficiently. In this work the authors propos an algorithm which uses an EKFAC-preconditioner with the standard Neumann series iterations to arrive at an accurate iHVP approximation. They numerical show that using their algorithm enhances the  performance of training data attribution methods.

**Questions:**

Please see my questions above,

**Ethical Concerns:**

["NO or VERY MINOR ethics concerns only"]

**Final Justification:**

I thank the authors for their detailed response. The response, in particular the scaling experiment, resolves most of my concerns. I will increase my score to 4. I need to add two points:

I still think the writing of the paper can be significantly improved. I encourage the authors to enhance the writing of the paper, in particular regarding the points I raised in the review.

As I mentioned in the review, I am not fully familiar with the literature of this subject, thus I could not comment confidently on the novelty of the paper. After reading other reviews, it seems that the paper is indeed novel, but I cannot confirm indepednently.

**Paper Formatting Concerns:**

No concerns

**Quality:**

2

**Strengths And Weaknesses:**

- The main problem that I have with the paper is that I am not convinced that the main bottleneck in influence functions is the accuracy of computing the inverse Hessian. Although the authors show some empirical benefits of their proposed algorithm that improvs the accuracy of  Hessian vector products, I'm not sure how much benefit this method has in real problems. I think the the paper studies a rather limited problem.

- The writing of this paper is not in an acceptable form. It is very hard to read the paper. The paper does not mathematically define many objects, even the actual problem definition is buried in the text. Acronyms are used without proper definition, etc. The writing needs to be significantly improved for the paper to be publishable. Even the proposed algorithm is not well presented. How is G_{EKFAC} calculated (precisely).

- The idea of the proposed algorithm is very simple. The authors observe that  Stochastic Neumann Series Iterations is equivalent to GD applied to the quadratic objective (4). Then they propose to replace GD with E-KFAC that takes into account the curvature of loss function. It is not very surprising that this methods performs better than the baseline.

- Does the modified NSI provably converge?

- How does the performance change as the models are scaled? A "scaling law" experiment would be beneficial.

- I am not very familiar with the prior work in this exact area, so I can not comment fully on the novelty of the proposed method for this problem.

- The actual choice of the preconditioner P seems to be arbitrary. How does this choice affect the performance? Is the preconditioner that you choose exactly the E-KFAC preconditioner?

- What happens if diagonal preconditioners are used instead of E-KFAC?

- How sensitive is the results to the choice of learning rate? can the authors provide a plot for one of their experiments that shows the loss at a fixed # of iterations, for different values of learning rate?

---

> ### Author Rebuttal · Authors · 2025-07-30
>
> Thank you for your detailed review! We can generally categorize your comments into two buckets: 1) concerns about the writing style of the paper and clarification questions about certain definitions and 2) doubts whether our contribution helps advance TDA. If there are any places that remain unclear after our discussion, please let us know - we think any ambiguities can be cleared up with very modest modifications to the manuscript.
>
> > The main problem that I have with the paper is that I am not convinced that the main bottleneck in influence functions is the accuracy of computing the inverse Hessian.
>
> In our experiments in Figure 2, we directly show that the iHVP is one of the bottlenecks -- across a range of settings, we find that improving the iHVP approximation in general leads to better Linear Datamodeling Score (LDS), holding everything else the same. This is especially true for convolution architectures.
>
> The fact that a better iHVP in general improves TDA outcomes is not a unique observation by us. In the extreme case, consider approximating the Hessian with the identity: Bae et. al. 2024 [1] and Choe et. al. 2024 [2] show that methods involving the Hessian (e.g. EKFAC influence) outperforms methods which do not use the Hessian (e.g. TracIn [7], Graddot).
>
> > I'm not sure how much benefit this method has in real problems. I think the the paper studies a rather limited problem.
>
> The core goal of TDA algorithms is to estimate how changes to the training dataset would affect the model’s predictions, without actually carrying out expensive retraining. Our evaluation metric LDS measures how well a TDA method correlates with ground-truth retraining, and is agnostic to the downstream task. This method of evaluation is common to many well-known TDA papers [1,2,3,4]. In this paper, we address the efficacy of TDA algorithm itself, rather than how this tool is applied to downstream tasks such as interpretability, data curation etc..
>
> > The writing of this paper is not in an acceptable form. It is very hard to read the paper.
> > The paper does not mathematically define many objects
>
> Thank you for this feedback. At the outset, when writing this paper, we recognized it would be difficult to fully convey all the necessary mathematical background in the main body since this paper is at the intersection of many topics: influence functions, unrolled differentiation, EKFAC preconditioning and stochastic Neumann series iterations (SNI). Given the page limit, we had to tradeoff whether to state in the main paper certain preliminaries that have been repeated before in related works (e.g. [8]), or whether to defer some of the preliminaries to Appendix B, and only reference them in the main paper. We do believe all the mathematical objects we discuss have been defined, if one reads the main paper in conjunction with Appendix B. We are happy to refer to Appendix B in a more specific way in the main paper and to pull forward more details about EKFAC.
>
> > even the actual problem definition is buried in the text.
>
> We are not quite sure what you mean by “actual problem definition”. If you mean that “the actual problem definition” is to compute an iHVP, lines 116-125 are dedicated to describing the LiSSA/SNI algorithm which computes the iHVP (line 125). If you mean “the actual problem definition” is to compute influence scores that correlate with ground-truth retraining, then you can find the evaluation metric LDS defined in Equation 6. Given the extra page for the camera ready, we are happy to improve the paragraph spacing and give certain equations their own lines.
>
> > Acronyms are used without proper definition, etc.
>
> Non-standard acronyms we use (e.g. SNI) are written in parentheses after the full term in the paper, with a full list of acronyms listed in Appendix A. Acronyms corresponding to algorithms used in TDA such as EKFAC and LiSSA are cited, and also its full name is listed in Appendix A. Could you let us know exactly where they are undefined?
>
> > Even the proposed algorithm is not well presented. How is $\\mathbf{G}\_{\\text{EKFAC}}$ calculated (precisely).
>
> The precise EKFAC Generalized Gauss Newton Hessian approximation $\\mathbf{G}\_{\\text{EKFAC}}$ is laid out in Appendix B.5 (Lines 1045-1087), computed in the same way as previous works [1,8]. We have not made any changes to this fairly standard computation procedure.  Unfortunately, our allowed rebuttal is limited in length, so we cannot lay it out here. However, to make things clear, we will add a line in the body of the paper that $\\mathbf{G}_{\\text{EKFAC}}$ is the block diagonal matrix, whose blocks correspond to layers in the neural network, and whose entries are $\\hat{\\mathbf{F}}_l$ described in Appendix B5.
>
> > It is not very surprising that this methods performs better than the baseline.
>
> We view this fact as a positive — our method is a principled one based on a large body of existing work and thus in principle is expected to work. Preconditioning gradient descent with EKFAC is approximate natural gradient descent [5, 6] where we approximate the Generalized Gauss Newton Hessian using EKFAC.
>
> > Does the modified NSI provably converge?
>
> Our algorithm is guaranteed to converge for a sufficiently small learning rate, an appropriate learning rate schedule, and the choice of $\\mathbf{P} := \\mathbf{G}_{\\text{EKFAC}}$ as is done in our paper.
>
> To see this, note that our update rule in Equation 5 is the mini-batch gradient descent rule for the quadratic:
>
> $$\\frac{1}{2} \\boldsymbol{\\theta}^\\top (\\mathbf{P} + \\tilde{\\lambda} \\mathbf{I})^{-1}(\\mathbf{G}+ \\lambda \\mathbf{I})\\boldsymbol{\\theta} - \\boldsymbol{\\theta}^\\top (\\mathbf{P} + \\tilde{\\lambda} \\mathbf{I})^{-1}\\nabla_{\\boldsymbol{\\theta}} f_{z_q} (\\boldsymbol{\\theta}^s)$$
>
> and for our choice of $\\mathbf{P} := \\mathbf{G}_{\\text{EKFAC}}$, $(\\mathbf{P} + \\tilde{\\lambda} \\mathbf{I})^{-1}(\\mathbf{G}+ \\lambda \\mathbf{I})$ is positive definite. We thus inherit properties of mini-batch gradient descent on strongly convex quadratics.
>
> > How does the performance change as the models are scaled? A "scaling law" experiment would be beneficial.
>
> We are at our character length, so for this question we point you to our response to scaling in our answer to reviewer ZrKT.
>
> > I am not very familiar with the prior work in this exact area, so I can not comment fully on the novelty of the proposed method for this problem.
>
> Thank you for your candid remarks. The novelty here is that if we were to compute influence using EKFAC, we might as well use this as a free preconditioner if we want to then apply an iterative algorithm such as SNI. The benefit of SNI is that as we apply more compute, the iHVP tends to improve -- it is a consistent estimator of the iHVP. In TDA, this combination of EKFAC influence and iterative methods has not been explored.
>
> > The actual choice of the preconditioner $\mathbf{P}$ seems to be arbitrary. How does this choice affect the performance? Is the preconditioner that you choose exactly the E-KFAC preconditioner? What happens if diagonal preconditioners are used instead of E-KFAC?
>
> Yes. The preconditioner we choose for all our experiments is exactly the EKFAC preconditioner, stated on line 184 and line 198. While in principle one can choose any preconditioner $\\mathbf{P}$ to minimize the quadratic objective in Equation 4, it makes a lot of sense to choose the EKFAC preconditioner for the following reason: To compute influence, many would by default adopt EKFAC-IF since it is a state-of-the-art method, which requires computing the EKFAC decomposition. Our insight is that we could repurpose this EKFAC decomposition as a preconditioner for Stochastic Neumann Series Iterations (SNI) to get a more accurate iHVP. Rather than recomputing a preconditioner, we might as well use the already computed EKFAC decomposition at no additional cost.
>
> We can consider what would happen if we use another preconditioner. First, note that by Equation 5, if we initialize the weights to zero, then after the first step, the solution obtained is the solution had we computed the iHVP using the (damped) preconditioner as the curvature matrix (we also explain this on line 189 onwards). Since EKFAC-based influence functions is a state-of-the-art TDA method, had we used another preconditioner (say a diagonal one), our iterative method would take many iterations just “catching up” to EKFAC-IF’s TDA performance.
>
> > How sensitive is the results to the choice of learning rate? can the authors provide a plot for one of their experiments that shows the loss at a fixed # of iterations, for different values of learning rate?
>
> While we cannot provide plots or links to external sites due to NeurIPS rules, we can comment on the general behavior. Fixing everything else (e.g. batch size to sample $\\tilde{\\mathbf{G}}_k$), since both SNI and our method provably converges for a small enough learning rate, in both cases, if one were to choose too small learning rate, more iterations would be needed to obtain an iHVP solution with the same loss as defined in Equation 4, and too large a learning rate causes divergence.
>
> ```
> References:
>
> [1]: Bae, Juhan et al. “Training Data Attribution via Approximate Unrolled Differentiation”.
> [2]: Choe, Sang Keun et al. “What is Your Data Worth to GPT? LLM-Scale Data Valuation with Influence Functions”.
> [3]: Park, Sung Min et al. “TRAK: Attributing Model Behavior at Scale".
> [4]: Mlodozeniec, Bruno et. al. "Influence Functions for Scalable Data Attribution in Diffusion Models".
> [5]: George, Thomas et. al. "Fast Approximate Natural Gradient Descent in a Kronecker-factored Eigenbasis".
> [6]: Amari, Shun-ichi et. al. "Natural Gradient Works Efficiently in Learning".
> [7]: Pruthi, Garima et. al. "Estimating Training Data Influence by Tracing Gradient Descent".
> [8]: Grosse, Roger et al. “Studying Large Language Model Generalization with Influence Functions”.
> ```

---

### Note · Authors · 2025-08-12

We sincerely thank all reviewers and Area Chairs for the time spent reviewing our paper.

**We highlight some of the positive comments the reviewers made:**

Reviewers thought our algorithm was “well-motivated” (reviewer ZrKT), had “practical impact” (reviewer CuEj). Reviewer ZrKT acknowledged that our algorithm is “a more accurate and cost-effective way to approximate inverse Hessian-vector products (iHVPs), which are central to influence-function-based TDA.”

Reviewers thought our method/derivations were “clear” (reviewer CuEj), “rigorous” (reviewer LxJj) and “principled” (reviewer CuEj). Reviewer CuEj thought our method was a "simple yet novel fusion of EKFAC and SNI that had not been explored for TDA.”

Reviewers thought the evaluations were “extensive” (reviewer LxJj) and “convincing” (reviewer CuEj). Reviewer Kz7b acknowledged “the authors show some empirical benefits of their proposed algorithm that improvs the accuracy of Hessian vector products”.

With respect to our ablation study in Section 6, reviewers thought it delivered “insightful analysis” (reviewer CuEj), "guiding future work” (reviewer CuEj), and that the analysis “validates the method” (reviewer LxJj), and “gives practitioners guidance on where to focus optimisation effort” (reviewer LxJj)

**We also addressed some common concerns of the reviewers in the rebuttal:**

- **Writing:**
    - We thank the reviewers for this feedback, and highlight that the paper is self-contained: details about SNI and EKFAC are in the extended preliminaries in Appendix B and currently referenced in main paper, but we are happy to move some parts forward as a result of the feedback.

- **Theoretical guarantees:**
    - Our method inherits favorable properties of mini-batch gradient descent on a strongly convex quadratic, and thus for a small enough learning rate and appropriate learning schedule, our method is guaranteed to converge.

- **The scale of the experiments:**
    - TDA methods attempt to approximate the outcomes of repeated retraining, which is the ground-truth we measure against and clearly intractable for large models. Our LDS experiments are similar in scale to recent TDA papers (e.g. Bae et. al. 2024, Choe et. al. 2024).
    - For the reason above, we cannot compute the LDS, but can check the values of the loss on the quadratic objective on larger models. We reran this check using ASTRA on a 1.5B parameter model GPT2-XL and noted the positive results to each reviewer.

---

### Decision · Program_Chairs · 2025-09-17

**Decision:**

Accept (poster)

**Comment:**

All four reviews leaned towards acceptance with one accept recommendation [CuEj] and three borderline accept recommendations [Kz7b,LxJj,ZrKT].

Reviewers appreciated several aspects of the work:

+ The algorithm was considered well motivated [ZrKT]
+ The method was considered simple but novel [CuEj]
+ Showing that influence scores can regain predictive utility was appreciated [ZrKT]
+ The method was considered to have practical impactt as a drop-in replacement [CuEj]
+ Improving generalisation of EKFAC-based methods to convolutional network architectures was appreciated [ZrKT]
+ Exploration of the impact of low curvature was appreciated [LxJj] and considered insightful [CuEj]
+ The method was considered principled with clear derivation and principled preconditioning [CuEj]; similarly, the mathematical derivation was considered rigorous [LxJj]
+ The evaluations were considered extensive [LxJj] and convincing [CuEj]
+ The accuracy improvement and cost-effectiveness were appreciated [ZrKT]


However, several concerns were raised, which the authors responded to at the rebuttal stage:

- Requiring second-order statistics and matrix-vector products was criticised and scalability to large-scale models was questionsd [ZrKT]; authors argued for necessity of second-order statistics and that strong TDA methods already compute second order information.
- One reviewer considered the studied problem limited [Kz7b]; authors argued they are addressing efficacy of the TDA algorithm itself.
- One reviewer was not convinced about what the main bottleneck in influence functions is [Kz7b]; authors discussed that their experiments show improving inverse Hessian approximation improves the TDA outcome.
- One reviewer considered the idea simple [Kz7b] and not surprising that it outperforms the baseline [Kz7b]; authors argued this to be a positive.
- Scalability to large models was considered unclear [LxJj,CuEj] and the largest model in experiments was considered small compared to scalability of EKFAC [LxJj]. Discussion on scalability to large architectures was desired [ZrKT,LxJj] and similarly, a "scaling law" experiment was desired [Kz7b]. Authors ran two larger experiments and commented about specific computation and memory costs of the steps.
- A further scaling question involved when eigenupdates dominate the cost; authors responded with some discussion.
- Lack of comparison to recent TDA methods was criticised [CuEj]; authors provided a comparison to two more methods and brief discussion of two others.
- Availability of theoretical guarantees/bounds on convergence or approximation error were desired [ZrKT,Kz7b]; authors discussed a convergence guarantee but noted they cannot guarantee the rate.
- Discussion of the impact of the preconditioner choice was desired, including if diagonal preconditions are used [Kz7b]; authors argued using EKFAC preconditioned to be natural, and argued using a different one would reduce performance.
- Discussion of sensitivity to learning rate was desired [Kz7b]; authors commented overly small learning rate would delay convergence and large would cause divergence.
- Results and discussion on settings where the method underperforms were requested [ZrKT]; authors provided some discussion stating they did not observe a reliable pattern.
- Dependence of performance on ensemble in some evaluations was criticised [LxJj]; authors argued they get improvements without ensemble too.
- The writing was considered dense [CuEj] and another reviewer considered the paper very hard to read and lacking definitions [Kz7b]; authors aimed to do some reorganisation of preliminary material and improve visibility of some equations and clarify some definitions.

After the rebuttals, [ZrKT] seemed to be mostly satisfied, [Kz7b] felt most of their concerns were resolved, whereas [LxJj] felt some of their concerns were addressed.

Overall, it seems if the authors incorporate well the planned revisions, and especially pay attention to readability which may still be a concern, the work may ultimately be at a sufficiently good state to be presented at NeurIPS.